# The flavonoid 4,4′-dimethoxychalcone promotes autophagy-dependent longevity across species

Didac Carmona-Gutierrez et al.[#]

Ageing constitutes the most important risk factor for all major chronic ailments, including malignant, cardiovascular and neurodegenerative diseases. However, behavioural and pharmacological interventions with feasible potential to promote health upon ageing remain rare. Here we report the identification of the flavonoid 4,4′-dimethoxychalcone (DMC) as a natural compound with anti-ageing properties. External DMC administration extends the lifespan of yeast, worms and flies, decelerates senescence of human cell cultures, and protects mice from prolonged myocardial ischaemia. Concomitantly, DMC induces autophagy, which is essential for its cytoprotective effects from yeast to mice. This pro-autophagic response induces a conserved systemic change in metabolism, operates independently of TORC1 signalling and depends on specific GATA transcription factors. Notably, we identify DMC in the plant *Angelica keiskei koidzumi*, to which longevity- and health-promoting effects are ascribed in Asian traditional medicine. In summary, we have identified and mechanistically characterised the conserved longevity-promoting effects of a natural anti-ageing drug.

The medical and socioeconomic advances experienced in developed countries over the last century have greatly extended life expectancy. However, health span has not increased at the same pace, resulting in the growing incidence and prevalence of age-related pathologies. Indeed, ageing remains the main risk factor for all major chronic maladies, including cardiovascular diseases, neurodegeneration and cancer[1]. Since the majority of ageing people are polymorbid, even considerable advances against a single age-related disease may only marginally improve health span. Therefore, tackling age-onset diseases by targeting their commonality, the ageing process itself, appears the most expedient approach. To date, only a few efficient dietary or pharmacological anti-ageing interventions exist; these include calorie restriction (the permanent reduction of total caloric intake without malnutrition) and administration of pro-longevity drugs like spermidine, rapamycin, metformin, NAD$^+$ precursors, or resveratrol[2,3]. Further approaches that are able to regress (or at least delay) the onset of pathogenic age-related decline are urgently needed.

Interestingly, epidemiological studies suggest that the regular consumption of polyphenol-rich foods may decrease the risk of many chronic conditions[4], and certain polyphenols—most prominently resveratrol—have been shown to extend life and/or health span in several model organisms ranging from yeast to mice[5]. Polyphenols are phytochemicals widely dispersed throughout the plant kingdom with manifold functions *in planta* ranging from pollinator attraction to pathogen and UV protection. Among them, the flavonoids represent the largest polyphenol subgroup and many of them show anti-inflammatory, anti-carcinogenic, anti-neurodegenerative and general cytoprotective properties[6,7]. However, reports specifically addressing the long-term effects of chemically defined flavonoids on ageing remain rare.

Most if not all behavioural, nutritional, pharmacological, and genetic manipulations that are known to extend lifespan stimulate macroautophagy (hereafter referred to as autophagy). In fact, autophagy seems to be a causal effector of these protective characteristics. For instance, the longevity drugs resveratrol, rapamycin, and spermidine, all lose their efficacy when autophagy is suppressed[2]. Autophagy is an intracellular recycling process, in which damaged or superfluous macromolecules and organelles are sequestered in two-membraned vesicles (autophagosomes) and then targeted to lysosomes for bulk degradation[8]. This facilitates the supply of recycled components for biosynthesis and thus contributes to cytoplasmic renewal and consequent cellular rejuvenation. Conversely, impairment or dysregulation of autophagic function results in age-related pathologies[9,10]. Altogether, autophagy is largely associated with cytoprotection and overall health.

Here we report the identification of the flavonoid 4,4′-dimethoxychalcone (DMC) as a natural autophagy inducer with phylogenetically conserved anti-ageing properties. We found that administration of DMC promotes cytoprotection and autophagy across species and that autophagy induction is required for the beneficial effects of this compound. Autophagy activation by DMC depends on specific GATA transcription factors, but not on the TORC1 kinase, a major regulatory instance of autophagy. This suggests synergistic potential with other anti-ageing interventions that do rely on TORC1 signalling.

## Results

**4,4′-dimethoxychalcone (DMC) promotes longevity across species**. In an effort to identify novel natural compounds with anti-ageing properties, we screened a library of 180 compounds representing different subclasses of flavonoids (Supplementary Table 1) for their ability to counteract age-related cellular demise. For this purpose, we monitored cellular health during yeast chronological ageing—an established model for the ageing of human post-mitotic cells[11–13]—in the presence of each of these flavonoids at a concentration of 50 μM. Using a high-throughput approach (Fig. 1a, Supplementary Fig. 1a–e), we determined in parallel (i) cellular membrane integrity (survival) by means of propidium iodide (PI) staining (Fig. 1b, Supplementary Fig. 1d), (ii) the clonogenic potential (outgrowth) of aged cells (Fig. 1b, Supplementary Fig. 1e), and (iii) the production of reactive oxygen species (ROS) detectable as the ROS-driven conversion of dihydroethidium to fluorescent ethidium (Fig. 1c). In each of these three independent assays, DMC emerged as a top cytoprotective hit. Upon further determining the concentration dependency of DMC's rescuing effect, we established the optimal dose in yeast to be at 100 μM (Supplementary Fig. 2a). DMC's potential to reduce chronological age-related cell death (as assessed by PI staining) was thereby comparable to that of several compounds previously reported as cytoprotective in ageing models. Precisely, DMC partly outperformed other polyphenols, including resveratrol and specific flavonoids (at 100 μM, as DMC), and yielded a similar protective capacity as rapamycin (at its optimal dosage of 40 nM) (Supplementary Fig. 2b). Confirming and extending our screening results, DMC treatment reduced the age-related increase in apoptotic and necrotic cell populations as determined by AnnexinV/PI co-staining, diminished the population of ROS-accumulating cells and promoted clonogenicity during ageing (Fig. 1d–f, Supplementary Fig. 2c, d).

We next examined whether DMC would act as an anti-ageing compound in two multicellular organisms, namely, the nematode *Caenorhabditis elegans* and the fruit fly *Drosophila melanogaster*. Remarkably, chronic DMC treatment (41.6 μM for worms, 200 μM for flies) prolonged the median lifespan of both model organisms by approximately 20% (Fig. 1g, h, Supplementary Fig. 3a–k, Supplementary Table 2). Of note, DMC did neither affect food intake (Supplementary Fig. 4a) nor fecundity (Supplementary Fig. 4b) of flies, and its longevity-extending effects were independent of the food composition (Supplementary Fig. 4c–e). DMC (50 μM) also largely prevented the senescence-mediated decrease of clonogenic survival in highly confluent human cells (U2OS osteosarcoma, HeLa cervical carcinoma, and H4 neuroblastoma cells) (Fig. 1i, j, Supplementary Fig. 3l). Altogether, these results suggest that DMC mediates phylogenetically conserved anti-ageing effects.

**Autophagy is required for the beneficial effects of DMC**. Most anti-ageing interventions depend on autophagy to exert their protective properties[2], and several flavonoids have been reported to stimulate autophagy[14]. Thus, we tested whether DMC would activate autophagy. Indeed, in yeast cells subjected to chronological ageing, DMC increased the autophagy-dependent redistribution of a GFP-Atg8 fusion protein towards vacuoles and its subsequent degradation to yield free GFP (Fig. 2a–c, Supplementary Fig. 5a–c). Moreover, DMC treatment stimulated the activity of Pho8ΔN60, a truncated form of alkaline phosphatase (ALP), the vacuolar delivery and activation of which depends on autophagy[15,16] (Fig. 2d, Supplementary Fig. S5d, e). In worms, DMC feeding for 48 h resulted in increased formation of autophagosomes as indicated by microscopic analysis of mCherry-fused LGG-1, the worm homologue of Atg8 (Fig. 2e, f). Moreover, brains of aged flies fed with DMC for 30 days exhibited reduced abundance of Ref(2)P, the fly homologue of human sequestosome-1 (SQSTM1/p62), suggesting increased autophagic flux, which generally leads to reduced abundance of Ref(2)P[17] (Fig. 2g, h). Similarly, DMC induced autophagic flux in several human cell

lines (U2OS, colorectal carcinoma HCT116, hepatoma HepG2), as determined by immunochemical detection of LC3 lipidation (LC3-II) and microscopic detection of cytoplasmic GFP-LC3 dots in the absence or presence of chloroquine, which stalls lysosomal autophagosome degradation (Fig. 2i, j, Supplementary Fig. 6a–h). In addition, DMC caused a reduction of SQSTM1/p62 levels in HCT116 and HepG2 cells (Supplementary Fig. 6i–k). Finally, DMC administered intraperitoneally efficiently triggered autophagic flux in the heart and liver from wild type (WT) C57BL/6 mice, as shown by increased LC3 lipidation, even after leupeptin treatment, a lysosomal protease inhibitor (Fig. 3a, b,

Supplementary Fig. 6l, m). In conclusion, DMC induces autophagic flux in all model systems tested from yeast to mammals.

We next examined a possible causality between the induction of autophagy and the protective effects of DMC. DMC reduced the infarction area in mice subjected to prolonged myocardial ischemia (with no reperfusion), an autophagy-associated and ageing-relevant setting[18–20], but only in wild type animals and not in mice with cardiac-specific autophagy deficiency (Fig. 3c, d, Supplementary Fig. 6n). Importantly, this dependence on autophagy also held true for the longevity phenotype. The knockout or knockdown of autophagy-related (ATG) genes,

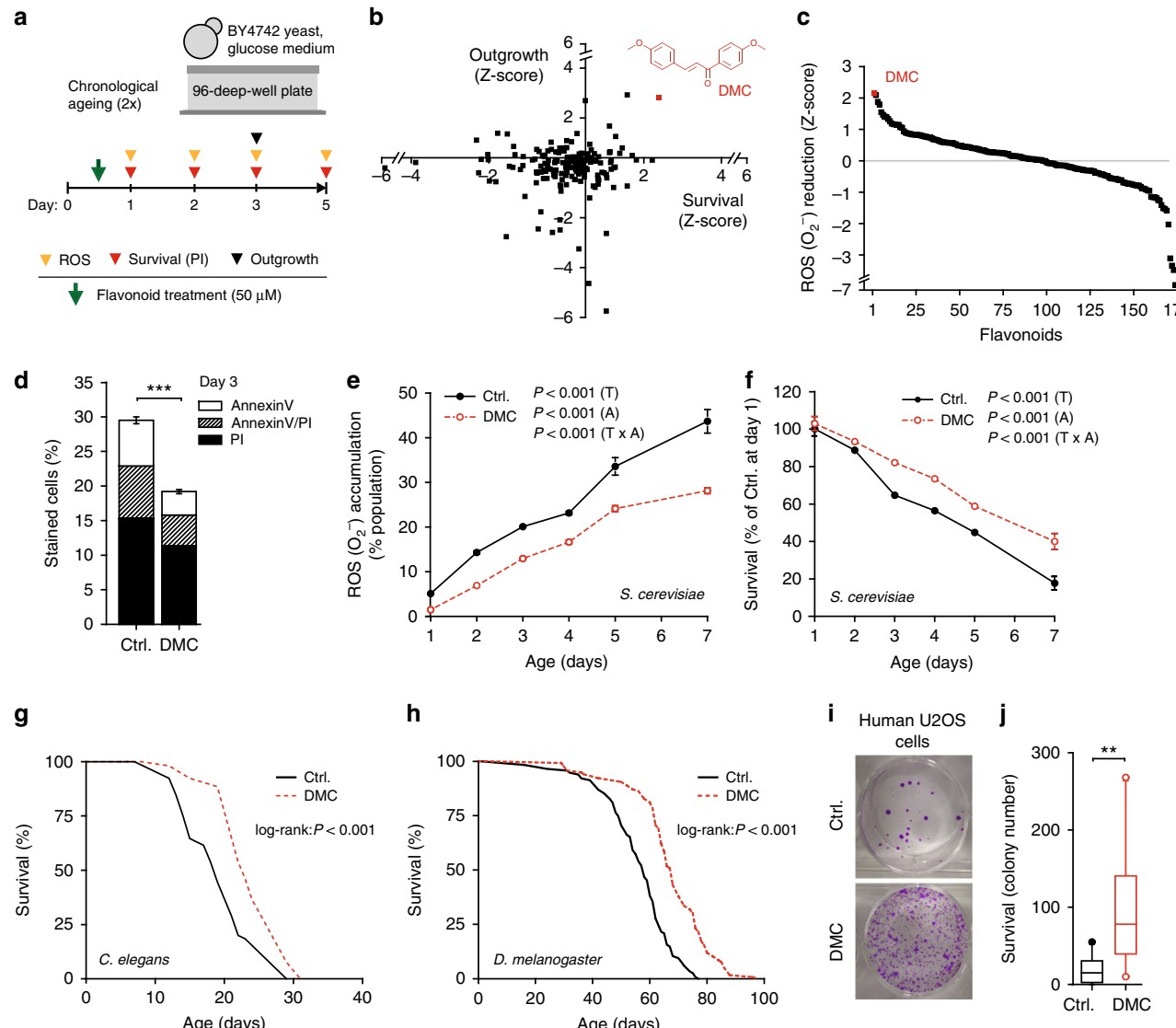

**Fig. 1** 4,4′-dimethoxychalcone promotes longevity in yeast, nematodes, flies and human cells. **a** Screening procedure for anti-ageing flavonoids in a yeast chronological ageing model. **b, c** Z-scores of AUCs obtained for each flavonoid during the yeast screen and for each assay performed: PI staining, outgrowth capacity (**b**), and DHE to ethidium conversion (**c**); data obtained in 1–2 independent runs with three replicates each. Each data point represents one flavonoid. The data point and the structure of 4,4′-dimethoxychalcone (DMC) are shown in red. **d–f** Phosphatidylserine externalisation and membrane dysintegrity (**d**), ROS production (**e**) and survival (**f**) of DMC-treated yeast cells (100 μM) at indicated time points of chronological ageing using AnnexinV/PI costaining (**d**), ***$P < 0.001$, $n = 18$, DHE to ethidium conversion, $n = 6$ (**e**) and clonogenicity, $n = 8$ (**f**) independent biological replicates; $P$-values represent T, treatment; A, age; T×A, interaction. **g, h** Survival of *C. elegans* (**g**) or *D. melanogaster* (**h**) during ageing with supplementation of food with DMC (41.6 μM for worms, 200 μM for flies). For other ageing replicates, see Supplementary Figure 2 (yeast) and 3 (nematodes, flies). **i, j** Replicative viability of DMC-treated (50 μM) U2OS cells. Representative images (**i**) and quantification (**j**) are shown, **$P = 0.0069$, $n = 11$ independent biological replicates. Data in (**d**) represent means ± SEM; box plots in (**j**) represent IQR (line at median) and whiskers 10–90 percentile. Significance in (**d, j**) was assessed by two-sided Student's *t*-test, in (**e, f**) by repeated measures two-way ANOVA. Source data for (**b–f, j**) are provided as a Source Data file

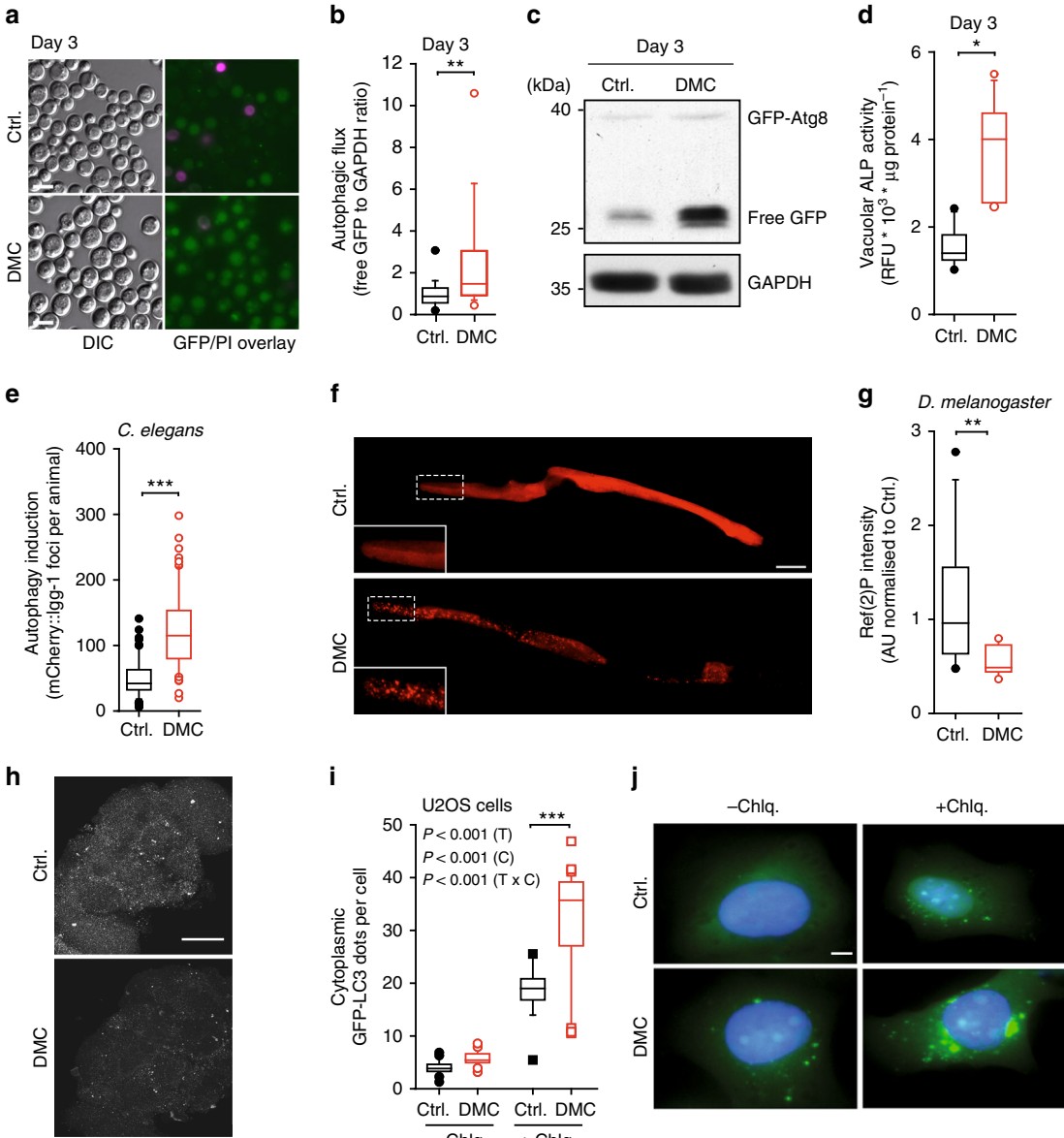

**Fig. 2** 4,4′-dimethoxychalcone induces autophagy across species. **a–d** Autophagic flux in DMC-treated (100 μM) yeast indicated by the vacuolar accumulation of GFP-Atg8 (green); a representative micrograph is shown in (**a**), where propidium iodide (PI) counterstaining served to visualise dead cells (magenta). Corresponding quantification was performed by analysing the free GFP/GAPDH ratio (**b**). **$P = 0.0024$, $n = 19$ (Ctrl.), 18 (DMC) independent biological replicates; **c** representative immunoblot. Autophagy measured via alkaline phosphatase (ALP) activity of Pho8ΔN60 strains (**d**). *$P = 0.0020$, $n = 11$ independent biological replicates. See Supplementary Figure 5d for other time points. **e**, **f** Quantification (**e**) and representative pictures (**f**) of mCherry-foci in the intestine cells of DMC-fed (41.6 μM) young adult worms expressing Pnhx-2::mCherry::lgg-1. ***$P < 0.0001$, $n = 61$ (Ctrl.), 65 (DMC) animals. Scale bar: 50 μm. **g**, **h** Immunofluorescence analysis (**g**) and representative pictures (**h**) of ref. [2] P-marked protein aggregates in fly brains after 30 days DMC feeding (200 μM). **$P = 0.0026$, $n = 12$ (Ctrl.), 11 (DMC) animals. Scale bar: 50 μm. Significance in (**b**, **e**, **g**) was assessed by two-sided Student's *t*-tests, in (**d**) by ANOVA/Bonferroni. **i**, **j** Videomicroscopic analysis (**i**) and representative pictures (**j**) of DMC-treated (50 μM) U2OS cells expressing GFP-LC3 with or without chloroquine, cell nuclei were stained with Hoechst 33342. Comparisons by two-way ANOVA (T, treatment; C, chloroquine; T × C, interaction) followed by Bonferroni-corrected simple main effects, ***$P < 0.0001$, $n = 22$ (Ctrl.-Chlq), 21 (DMC-Chlq), 18 (Ctrl. + Chlq), 30 (DMC + Chlq) independent biological replicates. Scale bar: 10 μm. Box plots represent IQR (line at median) and whiskers 10–90 percentile. Source data for (**b**, **d**, **e**, **g**, **i**) are provided as a Source Data file

which code for essential components of the autophagic machinery, abolished DMC-mediated lifespan extension in yeast, worms, and flies (Fig. 4, Supplementary Fig. 7a–h). Altogether, this indicates that autophagy induction is indeed necessary for the beneficial effects of DMC.

### DMC targets the yeast GATA transcription factor Gln3. Next, we asked via which pathway(s) DMC might promote autophagy.

DMC did neither impair proteasome activity nor induce endoplasmic reticulum stress (Supplementary Fig. 8a, b), thus ruling out two possible xenobiotic-mediated, pro-autophagic routes[21–23]. We then decided to examine the anti-ageing effects of DMC in several yeast mutants deficient for genes/proteins known to be involved in autophagic signalling[24] (Fig. 5a). The disruption of the GATA transcription factor GLN3[25] was the genetic intervention that most prominently abated DMC-mediated cytoprotection (Fig. 5a, b, Supplementary Fig. 8c, d). Of note, this phenotype

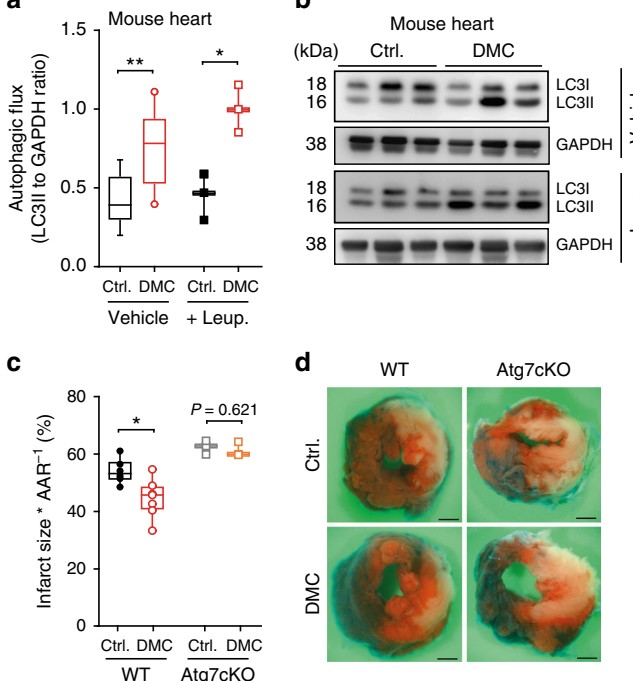

**Fig. 3** 4,4'-dimethoxychalcone promotes autophagy and cardioprotection in mice. **a**, **b** Autophagy induction in mouse heart tissue (**a**) determined by LC3 lipidation with leupeptin or vehicle injection after intraperitoneal injection of DMC (100 mg/kg) or DMSO (Ctrl.). n = 9 (Ctrl. Vehicle), 10 (DMC Vehicle), 3 (Leup.) animals, *P = 0.0105, **P = 0.0028; **b** representative immunoblot. **c**, **d** Infarction area per area at risk (AAR) (**c**) and representative images of left ventricular myocardial sections (scale bar: 1 mm) (**d**) after DMC treatment as in (**a**–**b**) followed by 3 h prolonged ischaemia in wild type (WT) and cardiac-specific Atg7 knockout mice (Atg7cKO). *P = 0.0134, n = 5 (Ctrl.), 6 (DMC), 3 (Atg7cKO) animals. Significance in (**a**, **c**) was assessed by two-sided Student's t-tests between Ctrl. and DMC. Box plots represent IQR (line at median) and whiskers 10–90 percentile. Source data for (**a**, **c**) are provided as a Source Data file

could be reversed by episomal Gln3 expression (Supplementary Fig. 8e, f), indicating that the observed effects were functionally linked to the deletion. DMC-mediated cytoprotection was also abrogated upon disruption of the Gln3 regulator protein phosphatase 2A (PP2A), either by deletion of its catalytic (PPH21/PPH22) or one of its regulatory (TPD3) subunits (Fig. 5a). The three other yeast GATA transcription factors (Gat1, Dal80 and Gzf3) failed to affect DMC treatment (Supplementary Fig. 9). Consistently, DMC treatment could not induce autophagy in the absence of Gln3 (Fig. 5c, d). This suggests a crucial and specific role of this GATA transcription factor in the execution of DMC effects.

Gln3 is involved in the co-regulation of general amino acid control[26], and interference with amino acid metabolism has been linked to lifespan extension and autophagy induction across species[27–29]. In accordance with an impact of DMC on Gln3 function, the metabolomic and proteomic profiles of DMC-treated yeast cells showed a pronounced influence on amino acid metabolism (Fig. 5e, Supplementary Fig. 10c, f, g). A similar metabolic repercussion was detected upon metabolomic analysis of heart and liver tissue of mice treated with DMC intraperitoneally (Supplementary Fig. 10a, b, d, e). Notably, the metabolic imprint of DMC-treated wild type yeast seems to be very similar to that of gln3 deletion mutants both with respect to the amino acid profile[30] (Fig. 5e) and the metabolome in general (Fig. 5f). This suggests that the impact of DMC may suppress Gln3 activity. To test this, we monitored the expression of the β-galactosidase-encoding lacZ gene placed under the control of the Gln3-inducible MEP2 promoter, which is specifically targeted by Gln3[31]. In wild type yeast cells, DMC treatment reduced lacZ activity to levels close to those observed in the GLN3 knockout strain (Fig. 5g). This further supports the possibility that DMC mediates its effect by inhibiting Gln3 activity. We reasoned that in this case, the genetic disruption of GLN3 should per se also promote cytoprotection and autophagy. Indeed, GLN3 deletion mutants exhibited decreased age-related cell death and higher autophagy levels than wild-type cells (Fig. 5b–d, Supplementary Fig. 8c, d). Thus, Gln3 deletion and DMC treatment have similar, epistatic effects on yeast cells with respect to autophagy induction and cytoprotection, supporting the idea that both act on the same pathway. Altogether, our data argue for an anti-autophagic role of

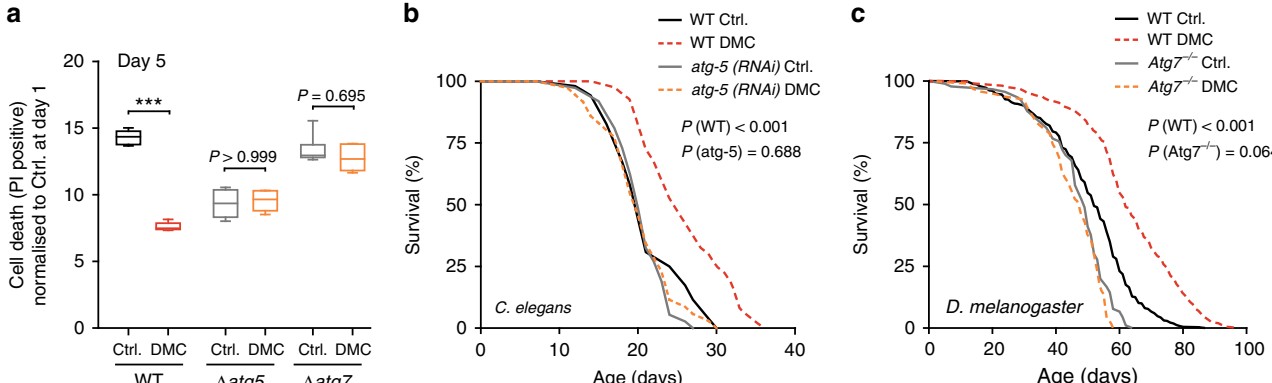

**Fig. 4** Autophagy induction is essential for 4,4'-dimethoxychalcone-mediated protection. **a** Survival of DMC-treated (100 μM) yeast wildtype (WT), ATG5-deficient, and ATG7-deficient mutant strains at day 5 of chronological ageing measured by PI staining. Data are normalised to the WT Ctrl. at day 1. Comparisons by two-way ANOVA with treatment and strain as independent variables, followed by Bonferroni-corrected simple main effects. ***P < 0.001, n = 6 independent biological replicates. Box plots represent IQR (line at median) and whiskers 10–90 percentile. Refer to Supplementary Figure 7 for other time points and other replicate experiments. Source data for (**a**) are provided as a Source Data file. **b**, **c** Survival of DMC-fed (41.6 μM) control and autophagy-deficient Atg5 RNAi nematodes (**b**) or female wildtype (WT) and Atg7-deficient (Atg7⁻/⁻) mutant flies (**c**) during ageing. Refer to Supplementary Figure 7 for other replicate experiments. P-values represent pairwise comparisons (Ctrl. vs. DMC) by log-rank analysis

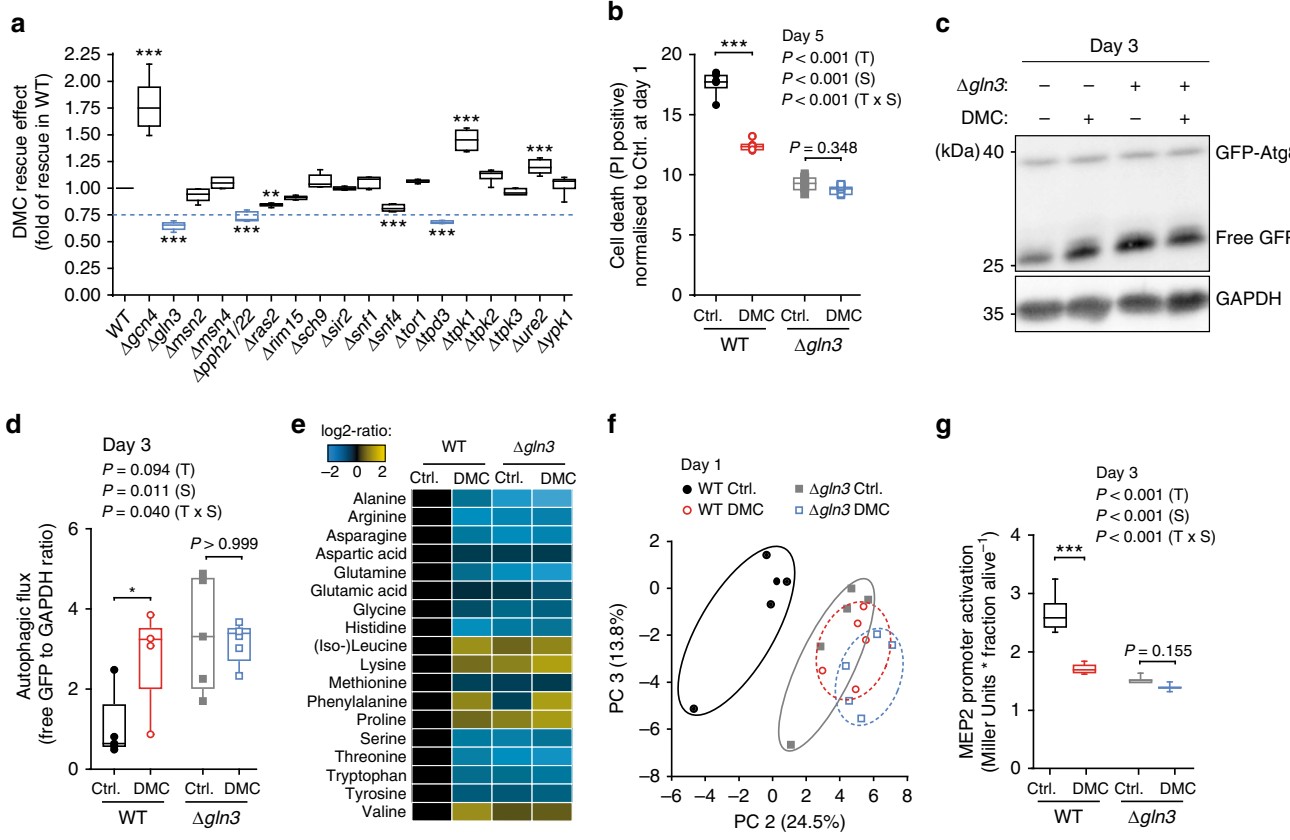

**Fig. 5** Gln3 is a functional target of 4,4′-dimethoxychalcone. **a** DMC-rescuing effect determined as the AUC of PI-positive cells (day 1–5) upon DMC treatment (100 μM) of yeast deletion mutants involved in autophagic signalling, normalised to the rescuing effect in wildtype (WT) cells. Hits below the threshold (0.75) are depicted in blue. Comparisons by ANOVA/Bonferroni. **P < 0.01, ***P < 0.001, n = 6 independent biological replicates. **b** Cell death at day 3 of chronological ageing of DMC-treated (100 μM) yeast wildtype (WT) and *GLN3*-deficient mutant strains determined by PI staining normalised to Ctrl. at day 1 (**b**), ***P < 0.001, n = 6 independent biological replicates. (**c**, **d**) Autophagy induction of DMC-treated (100 μM) yeast wildtype (WT) and *GLN3*-deficient mutant strains indicated by the free GFP to GAPDH ratio normalised to WT Ctrl. (**d**), *P = 0.0242, n = 5 independent biological replicates; representative immunoblot in (**c**). **e**, **f** Changes in intracellular amino acid concentrations (**e**) and principal component analysis of yeast metabolites (**f**) 24 h after DMC-treatment (100 μM) in wildtype (WT) and *GLN3*-deficient yeast cells. **g** Gln3-dependent MEP2 expression using a $P_{MEP2}$-LacZ reporter at day 3 after DMC-treatment (100 μM) in wildtype (WT) and *GLN3*-deficient yeast cells. **P = 0.0056, n = 12 (WT) n = 9 (Δ*gln3*) independent biological replicates; Comparisons in (**b**, **d**, **g**) by two-way ANOVA (T, treatment; S, strain; T × S, interaction) followed by Bonferroni-corrected simple main effects. Box plots represent IQR (line at median) and whiskers 10–90 percentile. Source data for (**a**, **b**, **d**–**g**) are provided as a Source Data file

Gln3 during chronological ageing that can be suppressed by DMC.

**DMC acts independently of TORC1.** We next asked how DMC might target Gln3 activity. The classical model of Gln3 cytoplasmic-nuclear translocation and activation involves TORC1-dependent negative regulation[32]. Thus, TORC1 inhibition, which is consistently connected to longevity[33], actually stimulates Gln3 activity and does not suppress it as DMC. Accordingly, treatment with rapamycin, an inhibitor of the TORC1 kinase complex with well-established anti-ageing effects, promoted activation of the Gln3-inducible MEP2 promoter, and this effect was completely lost upon *GLN3* deletion as previously described[31] (Supplementary Fig. 11a). Intriguingly, however, rapamycin continued to mediate both cytoprotection and autophagy induction under these GLN3-defective conditions. Deletion of Gln3 did neither preclude the reduction of age-dependent cell death (as determined by PI staining) nor the stimulation of autophagic flux (as determined by GFP liberation) induced by rapamycin (Fig. 6a–c, Supplementary Fig. 11b, c). These results indicate that DMC and rapamycin mediate cytoprotection via independent pathways. Consequently, the combinatorial treatment with both agents, which consistently reduced Gln3 activity compared to rapamycin supplementation alone (Supplementary Fig. 11d), resulted in additive cytoprotective effects (Supplementary Fig. 11e). Thus, DMC seems to operate independently from TORC1. Indeed, neither deletion of the main TORC1 component TOR1 nor YPK1, a downstream target kinase of TORC2 involved in the positive regulation of autophagy during amino acid starvation[29], influenced DMC effects (Fig. 5a). In line, as opposed to rapamycin, DMC did not influence ribosomal protein S6 (Rps6) phosphorylation, a downstream marker of TORC1 activity (Fig. 6d; Supplementary Fig. 11f). These data suggest that DMC regulates Gln3 activity in a manner that differs from the classical TORC1-dependent route.

We further unveiled a functional interaction of DMC with the Gln3 regulator PP2A. Its genetic inactivation by deletion of both its functionally redundant catalytic subunits (PPH21, PPH22) receded DMC effects (Fig. 5a, Supplementary Fig. 11g). These catalytic subunits can form distinct complexes with multiple regulatory/specificity subunits relevant for Gln3 regulation. One of these complexes is formed with the scaffold module Tpd3, the deletion of which we found to revert DMC-mediated effects (Fig. 5a, Supplementary Figure 11g), similar to PPH21/22 disruption. Notably, this specific complex—as opposed to the

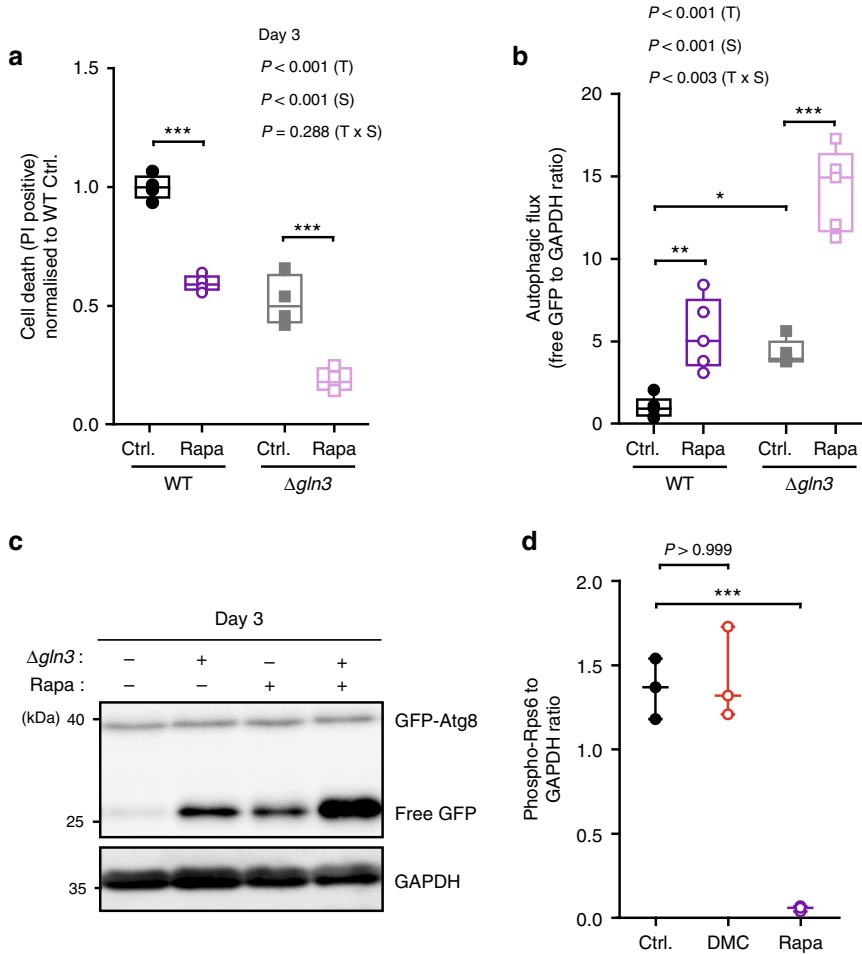

**Fig. 6** Rapamycin and 4,4′-dimethoxychalcone follow independent routes of cytoprotection. **a–c** Cell death assessed by PI staining and flow cytometry normalised to WT Ctrl. (**a**) and autophagy induction indicated by the free GFP to GAPDH ratio normalised to WT Ctrl. **b**, **c** of *S. cerevisiae* wildtype and *GLN3*-deficient mutant strains after treatment with 40 nM rapamycin (Rapa) at day 3 of chronological ageing. Comparisons in (**a**, **b**) by two-way ANOVA (T, treatment; S, strain; TxS, interaction) followed by Bonferroni-corrected simple main effects. $*P < 0.0483$, $**P < 0.0055$, $***P < 0.001$, $n = 4$ (**a**), 5 (**b**) independent biological replicates. (**d**) Rps6 S232/S233 phosphorylation of *S. cerevisiae* wild-type cells after 6 h of DMC (100 μM) or 40 nM rapamycin (Rapa) treatment as determined by immunoblotting. Comparisons by ANOVA/Bonferroni. $***P < 0.001$, $n = 3$ independent biological replicates. Box plots represent IQR (line at median) and whiskers 10–90 percentile. Source data for (**a**, **b**, **d**) are provided as a Source Data file

one with Tap42—is rapamycin-insensitive[34,35], supporting the idea that TORC1 is not involved in DMC effects. In aggregate, our data suggest that DMC functionally targets Gln3 in a TORC1-independent manner.

**GATA transcription factors are conserved DMC effectors.** Finally, we tested whether the described dependence of DMC on yeast Gln3 is phylogenetically conserved. Indeed, silencing of the *C. elegans* GATA transcription factor *elt-1*, a Gln3 homolog, precluded both DMC-mediated lifespan extension and autophagy induction in worms (Fig. 7a–c, Supplementary Fig. 12a, b). Similarly, *elt-1* knockout animals treated with DMC did not show improved survival during ageing (Supplementary Fig. 12c–e). Intriguingly, in human U2OS cells, siRNA-mediated silencing of GATA2 (and to a lower extent also that of GATA3, but not that of other Gln3 homologues), precluded autophagy induction by DMC (Fig. 7d, e, Supplementary Fig. 13). In contrast, rapamycin-induced autophagy remained unaltered upon GATA2 silencing (Supplementary Fig. 14a), again supporting the concept that DMC and rapamycin ignite independent pathways across species. In fact, DMC—as opposed to rapamycin—did not inhibit

TORC1 signalling as determined by sustained S6K1 phosphorylation levels (Supplementary Fig. 14b). In aggregate, these results indicate a conserved role for specific Gln3-like GATA transcription factors in DMC-mediated autophagy and lifespan extension.

## Discussion

While the beneficial effects of certain behavioral and dietary strategies (especially calorie restriction) are uncontestable[2], most individuals have difficulties to strictly and permanently adhere to them. This has encouraged the search for potential pharmacological alternatives. The present work identifies the flavonoid 4,4′-dimethoxychalcone (DMC) as an anti-ageing compound with cardioprotective effects in mice and the potential to promote longevity across species. This echoes prior studies reporting that chalcones possess a wide—though poorly defined—range of biological activities relevant to human health[6]. So far, DMC has been only reported to mediate a mild anti-malarial (IC$_{50}$ 21.7 μM)[36] activity and to inhibit the proliferation of human K562 leukaemia cell lines (IC$_{50}$ 15 μM)[37]. To our knowledge, there is no natural source of DMC known to date. Intriguingly,

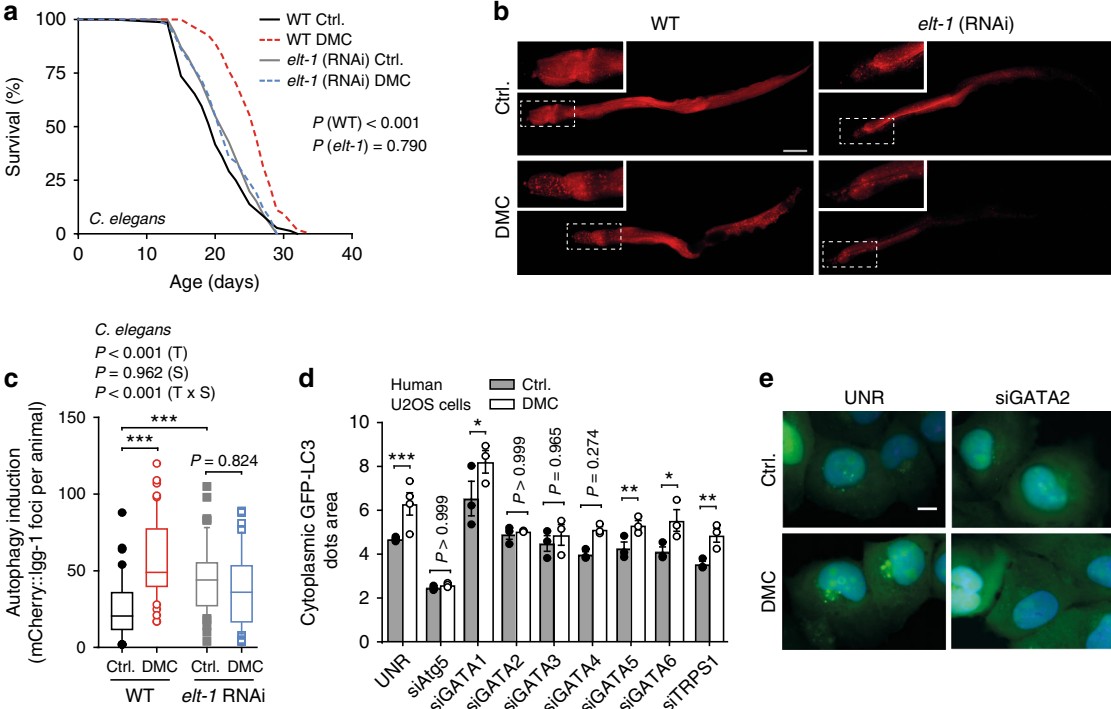

**Fig. 7** GATA transcription factors are phylogenetically conserved effectors of 4,4'-dimethoxychalcone. **a** Survival of DMC-fed (41.6 µM) control and GATA transcription factor-deficient *elt-1* RNAi nematodes. *P*-values represent pairwise comparisons by log-rank analysis. Refer to Supplementary Figure 12 for other replicate experiments. **b**, **c** Representative fluorescence pictures (**b**) and quantification (**c**) of mCherry-positive foci in the intestine cells of DMC-fed (41.6 µM) or untreated control or GATA-factor-deficient *elt-1* (RNAi) nematodes expressing $P_{nhx-2}$::mCherry::*lgg-1* reflective of autophagosome generation. Comparisons by two-way ANOVA (T, treatment; S, strain; TxS, interaction) followed by Bonferroni-corrected simple main effects. ***$P < 0.001$, $n = 44$ (WT Ctrl.), 53 (WT DMC), 57 (*elt-1* Ctrl.), 51 (*elt-1* DMC) animals. Scale bar: 50 µm. Box plots represent IQR (line at median) and whiskers 10–90 percentile. **d**, **e** Autophagy induction in DMC-treated (50 µM) human U2OS cells with siRNA against an unrelated sequence (UNR) or GATA transcription factors as determined via videomicroscopy of cells expressing GFP-LC3 (**d**). Cell nuclei were stained with Hoechst 33342. Data shown as means of three different siRNA constructs ± SEM. Comparisons by ANOVA with Tukey correction. *$P < 0.05$, **$P < 0.001$, ***$P < 0.001$, $n = 4$ (UNR), 3 (others) independent biological replicates. Representative images are shown in (**e**). Scale bar: 10 µm. Source data for (**c**, **d**) are provided as a Source Data file

we could detect DMC in the stipes and leaves (but not in the roots) of the chalcone-rich plant *Angelica keiskei koidzumi* (commonly known under the Japanese name of Ashitaba), to which longevity- and health-promoting effects are attributed in Asian folk medicine (Supplementary Fig. 15a, b). This fuels the expectation that DMC may be therapeutically applicable in humans. Incidentally, DMC seems to be well tolerated in mice with no apparent side effects or toxicity, at least up to a dose level of 2000 mg/kg *per os* during an observation time of 14 days[38]. Moreover, we could detect DMC in the blood plasma of middle-aged mice that were fed with chow containing 0.25% DMC for 7 days, suggesting that orally administered DMC becomes bioavailable in mammals (Supplementary Fig. 15c).

Our data further demonstrate that the protective effects of DMC are mediated by autophagy induction. Intriguingly, the anti-ageing capacity of most interventions—independently of their upstream targets—seems to converge in autophagy induction. Still, many flavonoids have antioxidant potential that might mediate some degree of acute cytoprotection separately from autophagy. For instance, DMC protected mice against hepatotoxicity induced by acute ethanol intoxication, as measured by decreased serum alanine aminotransferase (ALT) activity, in both wild type and whole body ATG4B knockout animals (Supplementary Fig. 6o). Along similar lines, DMC slightly reduced cell death during the early phase of chronological ageing in different yeast strains, were they autophagy-competent or not (Supplementary Fig. 7h). However, at later time points, the anti-ageing effects of DMC were largely lost in autophagy-deficient yeast

strains. In fact, the antioxidant capacity does not solely determine the long-term effects of cytoprotective flavonoids, as indicated by the fact that several flavonoids with powerful antioxidant activity[39] failed to mediate anti-ageing effects in our initial screen (Supplementary Fig. 2e). Accordingly, other studies have shed doubts on the exclusive relationship between the health-promoting and antioxidant properties of flavonoids[40].

DMC promotes autophagy via a pathway that involves specific GATA transcription factors. In yeast, our results suggest that during chronological ageing, the GATA transcription factor Gln3 limits survival by exerting an anti-autophagic role, and that this can be reverted by DMC. While a previous genetic high-throughput study, in which Δ*gln3* cells were found to be long-lived[41], supports this notion, the role of Gln3 in autophagy seems to be more controversial. In discrepancy with our data, Gln3 has previously been shown to promote *ATG14* expression upon nitrogen starvation in the logarithmic phase, raising the presumption that it might have a general pro-autophagic role[42]. However, this work did not measure actual autophagic flux in *gln3*-deficient strains, and ATG protein expression levels alone are not an accurate means to monitor autophagy[15,43]. In addition, a more recent study has challenged these observations by not finding any changes in *ATG14* transcription or autophagy induction in *gln3* deletion mutants subjected to nitrogen starvation, although other ATGs seemed to be upregulated[44]. More importantly, this latter report also showed that under *basal* autophagic conditions in the logarithmic phase, the deletion of *GLN3* (but not *GAT1*) actually promoted accumulation of *ATG8*

and *ATG29* mRNAs (*ATG14* remained unaffected).[44] The authors of that study concluded that Gln3 must have a direct or indirect role in the repression of some ATG genes when autophagy functions at basal/physiological levels[44]. Accordingly, we observed increased expression levels of different ATG proteins in ageing Δ*gln3* cells (Supplementary Fig. 16). While under acute autophagy induction, e.g. upon nitrogen starvation, Gln3 might positively contribute to the expression of distinct ATGs[44], our data and the recent literature[41,44] are consistent with a role for Gln3 in restraining cytoprotective autophagy operating at a basal activity, at least during physiological chronological ageing conditions. We thus surmise that, by interfering with the autophagy-repressive activity of Gln3, DMC can increase autophagic flux thus conferring cytoprotection upon ageing.

Both Gln3 activation[32] and lifespan extension[33] are associated to inhibition of TORC1. However, we show here that these two outcomes are correlative, i.e. that Gln3 activation as it occurs during TORC1 inhibition is not causal for cytoprotection. Similarly, our results support the notion that Gln3 is not required for rapamycin-mediated autophagy induction, as *GLN3* deletion and rapamycin treatment induce autophagy in an additive manner. These data suggest, on the one hand, that DMC regulates Gln3 activity TORC1-independently. Indeed, recent evidence indicates that such non-canonical mechanisms of Gln3 regulation may exist[45–47]. On the other hand, it establishes synergistic potential with other anti-ageing interventions that act on the TORC1 inhibitory pathway. As a proof of principle, we show that DMC and rapamycin (a specific TORC1 inhibitor) exert additive cytoprotective effects.

Several GATA transcription factors have been connected to lifespan control[48] in higher eukaryotes, and we can show that—as in yeast—DMC effects depend on specific GATA transcription factors in nematodes and human cell cultures. It should be noted that in worms, silencing of the GATA transcription factor *elt-1* did not *per se* promote organismal lifespan extension (as opposed to yeast Gln3), although it did increase autophagy in intestinal cells. This might be due to tissue-restricted effects of elt-1 RNAi or due to a higher complexity of longevity regulation in higher eukaryotes through interactions among several GATA family members[49]. In mammalian cells, the role of GATA transcription factors in autophagy is only starting to be elucidated with differing outcomes depending on the specific factor. Only two reports have addressed this issue[50,51]. On the one hand, GATA1 has been shown to promote autophagosome formation and transcription of genes encoding ATG components in human erythroblasts, although the effects on different ATGs were rather heterogeneous[50]. On the other hand, GATA4 seems to inhibit doxorubicin-mediated autophagy in rat cardiomyocytes[51]. Our data identifies GATA2 as a further family member with anti-autophagic capacity, at least under the tested conditions.

In the last century, life expectancy has increased globally, resulting in a demographic transformation characterised by an increased ageing population. This has greatly boosted the incidence and prevalence of late onset afflictions, which represents a pressing socioeconomic concern. Thus, it is critical to devise feasible anti-ageing interventions. The present work establishes the potential of DMC as a pharmacological instrument against ageing and age-related diseases. Future studies must explore whether DMC and/or its chemically defined derivatives can be advantageously used in humans as well.

## Methods

**Reagents**. The following reagents were purchased from the indicated suppliers: FITC-labeled Annexin V (Roche Applied Science [11828681001]), diyhdroethidium (DHE, Sigma-Aldrich, [D7008]), propidium iodide (PI, Sigma-Aldrich [P4170]), 4,4′-dimethoxychalcone (DMC; ABCR, [AB179040]);

Extrasynthese, [1295]; Sigma-Aldrich, [S617237]), rapamycin (LC laboratories, [R-5000]), resveratrol (Sigma-Aldrich [R5010]). For the initial yeast screen, all flavonoids (including 4,4′-dimethoxychalcone) were purchased from Extrasynthese; for a complete list including the name, subclass and article number of each flavonoid please refer to Supplementary Table 1. Flavonoid stock solutions were always freshly prepared in DMSO prior to treatment. Acetonitrile (Chromasolv), formic acid (puriss), ethylacetate (Chromasolv) were purchased from Sigma Aldrich (St Lois, USA). Water was purified by MilliQ system (<18.2 MΩcm, Merck, Darmstadt, Germany).

**Yeast strains and plasmids**. Experiments were carried out in BY4742 (MATα *his3Δ1 leu2Δ0 lys2Δ0 ura3Δ0*) and respective null mutants, obtained from Euroscarf or self-generated (Supplementary Table 3) as described below. To monitor subcellular localisation of endogenous Atg8, a previously reported strain expressing an EGFP–Atg8 fusion protein was used[52]. Atg protein expression was determined using chromosomal 6(HA)-tagged strains, generated using pYM17 as a template[53]. All strains were grown on SC medium containing 0.17% yeast nitrogen base (BD Diagnostics, Schwechat, Austria), 0.5% $(NH_4)_2SO_4$, 30 mg/l of all amino acids (except 80 mg/l histidine and 200 mg/l leucine), 30 mg/l adenine and 320 mg/l uracil, with 2% glucose (SCD). For complementation experiments with Gln3, i.e. for episomal Gln3 expression, cells were grown on SCD as described below and upon flavonoid treatment galactose was added to a final concentration of 0.1% to induce expression. Successful expression was verified via western blot; note that the double band obtained (Supplementary Fig. 8f) is characteristic for Gln3[54].

**Yeast chronological ageing experiments**. Chronological ageing experiments were carried out in 96-deepwell plates (Bel-Art Products, USA), sealed with gas-permeable adhesive membranes (Excel Scientific) and lids. Therefore, 500 µl of fresh media were inoculated with overnight cultures from cellular material of a 4-day-incubated YEPD plate to an $OD_{600}$ of 0.1 (~1.10^6 cells ml^−1). Thereby, the wells at the outer rims of the deepwell plates were not inoculated, since they are prone to dry out in the course of ageing; instead they were loaded with water. Next, cells were grown to an $OD_{600}$ of ~0.2 and subsequently supplemented with either the indicated concentration of freshly DMSO-diluted flavonoid or DMSO alone (control), both at a final concentration of 0.2% DMSO. Aliquots were taken out to perform different viability stainings (PI, DHE, AnnexinV/PI), survival tests (plating, regrowth capacity)[13,55], and/or autophagy assays (EGFP-Atg8, ALP) at indicated time points (Fig. 1a).

For the initial yeast screen, each flavonoid was supplemented at a concentration of 50 µM. After identification of DMC as a top cytoprotective hit, we determined the optimal dose for treatment in yeast at 100 µM, at least under the applied conditions. This was the lowest concentration exerting the highest rescue effect (Supplementary Fig. 2a).

One of the major causes of yeast chronological ageing might be the excessive production of acetic acid[56]. Consistently, yeast lifespan can be prolonged by alkalinization of the medium[52]. Of note, DMC treatment did not alter the pH of the medium (Supplementary Fig. 2f), demonstrating that the cytoprotective effects of DMC upon yeast chronological ageing are pH-independent.

If not otherwise stated, representative ageing experiments are shown with at least six independent biological replicates aged at the same time. Experiments were performed at least three times in total with similar outcome, except during the screening procedure, where each flavonoid was tested in three to six independent samples for each ageing experiment.

**Assays for cell death markers in yeast**. For the initial yeast screen, PI (necrosis) and DHE (superoxide anione $O_2^-$ production) staining was performed as follows: ~1 × $10^7$ cells were harvested by centrifugation for 5 min at 2,700 × g and stained for 5 min with propidium iodide (100 ng/ml in PBS pH 7.4) or dihydroethidium (2.5 µg/ml in PBS pH 7.4), respectively. Cells were pelleted again for 5 min at 2700 × g and resuspended in PBS. Relative fluorescence units were determined using a fluorescence reader (Tecan, GeniusPRO) and normalised to the $OD_{600}$ of each sample. Then, the area under the curve (AUC) for all monitored days throughout chronological ageing was calculated and a Z-score computed for the results with PI and DHE, respectively. The results obtained in these high-throughput assays positively correlate with the corresponding low scale experiments (Supplementary Fig. 1a–c). Of note, an unstained sample of each well was tested at day 1 of chronological ageing to take account for possible intrinsic fluorescence properties of any given flavonoid that might interfere with our fluorescence-based assays. For the viability assay using outgrowth capacity, which has been previously used to determine ageing-associated viability[41,57], aliquots (9 µl) were taken at day 3 of chronological ageing to inoculate 171 µl fresh SCD media in 96-well-plates. Cultures (total 180 µl) were grown at 28 °C, 1000 rpm (motor speed) and $OD_{600}$ was measured at the point of inoculation and 10 h thereafter using a plate reader (Tecan, GeniusPRO). Outgrowth was defined as the difference between the measured $OD_{600}$ at the time of inoculation and after 10 h of growth [$OD_{600(10h)}$- $OD_{600(inoculation)}$], and then normalised to the DMSO-treated control; finally, a Z-score was computed. As in the other high-throughput assays (see above), a positive correlation between outgrowth capacity and viability was established (Supplementary Fig. 1b).

Assays for apoptosis/necrosis (AnnexinV/PI co-staining) upon DMC treatment were quantified by flow cytometry (BD LSRII Fortessa, BD Biosciences). Briefly, ~$1 \times 10^7$ cells were harvested by centrifugation for 5 min at $2700 \times g$ and washed once with water and once with buffer B + S (35 mM potassium phosphate buffer, pH 6.8, 0.5 mM $MgCl_2$, 1.2 M sorbitol). To obtain spheroplasts, cells were resuspended in 330 µl buffer B + S, 15 µl glucuronidase/arylsulfatase (Sigma-Aldrich, [BGALA-RO]) and 3 µl lyticase (1000 U/ml, Sigma-Aldrich [L2524]) were added, and cells were incubated at 28 °C for 30 min. Spheroplasts were pelleted at 500 g for 2 min and carefully washed once with buffer B + 0.6 M sorbitol. Then spheroplasts were resuspended in 30 µl incubation buffer (10 mM HEPES pH 7.4, 140 mM NaCl, 5 mM $CaCl_2$, 0.6 M sorbitol) and stained for 20 min by addition of 3 µl AnnexinV-FLUOS (Sigma-Aldrich [11828681001]) and 3 µl PI (100 µg/ml). For flow cytometry, a total of 30,000 cells per sample were evaluated using BD FACSDiva software (BD Biosciences). ROS accumulating cells were quantified evaluated by DHE staining (see above) and subsequent flow cytometry and analysis of 30,000 cells. For clonogenic survival plating, cell counts of DMC-treated cultures and controls were measured using a CASY cell counter (Schärfe System GmbH), and 500 cells were plated on YEPD agar plates and incubated for two days at 28 °C to allow colony formation. The colony-forming units (CFUs) were analysed using an automated colony counter (LemnaTech). For each strain, the CFUs determined for the control cultures were set to 100%, and the survival of the respective DMC-treated cultures was calculated relative to the corresponding control culture.

**Plasmid construction and yeast knockout generation.** Single and double-mutant strains were generated according to a reported method by either employing a gene-specific URA3-knockout cassette, amplified by PCR using pUG72 as a template[58] or using the natNT2 or hphNT1 cassettes of pFA6a–natNT2 and pFA6a–hphNT1, respectively[53]. Correct gene deletion was verified by PCR with corresponding control primers and further checked for consistent auxotrophies. All primers used are listed in Supplementary Table 4. The plasmid [pESC-His-6(HA)] was constructed by digestion with SacI/ClaI and ligation with the 6(HA) fragment amplified from pYM17 using primers ClaI_6HA_f and SacI_6HA_r. GLN3 was amplified from genomic DNA from BY4742, using primers GLN3_F (NotI) and GLN3_R (ClaI) and cloned into the resulting vector using NotI/ClaI restriction sites (see Supplementary Table 5). Notably, at least three different clones of each generated mutant were tested for similar response to PI staining during ageing to rule out clonogenic variation.

**Yeast autophagy measurements.** Autophagy was examined according to published methods by determining either GFP liberation via immunoblot analysis (see Immunoblotting section) or vacuolar localization of Atg8 through fluorescence microscopy in cells expressing an EGFP–Atg8 fusion protein[52]. In addition, ALP activity[16] was assessed using corresponding Pho8ΔN60 cells transformed with and selected for stable insertion of pTN9 HindIII fragment[59]: ~$3 \times 10^7$ cells were harvested by centrifugation at $2700 \times g$ for 5 min and washed once with 500 µl $H_2O$. Cells were resuspended in 200 µl ice-cold Assay buffer (250 mM Tris/HCl pH 9.0, 10 mM $MgSO_4$, 10 µM $ZnSO_4$) and transferred to a pre-cooled reaction tube with 100 µl acid-washed glass beads. Cells were disrupted in a beadmill (Mini-Bead-beater) in a pre-cooled metal reaction tube rack for $3 \times 45$ s with 30 s intervals in between. Cell debris was removed by centrifugation at $10,000 \times g$ at 4 °C for 10 min and the supernatant carefully transferred to fresh pre-cooled tubes. Protein concentration was determined using a Bradford protein assay (Bio-Rad [5000006]). Cell extracts corresponding to 1–1.5 µg protein were added to a final volume of 550 µl assay buffer (room temperature) and the reaction was started by addition of 50 µl α-naphtylphosphate (55 mM in assay buffer, pH 9.0). After 20 min at room temperature, the reaction was stopped by addition of 200 µl stopping buffer (2 M glycin/NaOH pH 11.0). Two hundred microlitres of the reaction mix was measured in black 96-well plates in a platereader (Tecan, GeniusPRO) at ex: 345 nm, em: 472 nm. To correct for intrinsic (background) ALP activity, control cultures (without pTN9) were simultaneously processed and ALP activity subtracted.

**Yeast TORC1 activity.** Cells were grown to an $OD_{600}$ of 0.2 and treated with either 100 µM DMC or 40 nM rapamycin for 6 h. Three $OD_{600}$ units were harvested and proteins extracted (see Immunoblotting section). Rps6 phosphorylation at serine 232 and 233 was detected with a phospho-S6 ribosomal protein (Ser235/236) antibody[60] (Cell Signaling [#2211 S], rabbit, 1:1000) and phosphorylation levels normalised to GAPDH. As controls, stationary phase cultures of wild type and rps6aS232A,S233A Δrps6b strains (kind gift from Dr. Tarek Moustafa) were refed with fresh SCD medium for 1 h before harvest.

**Yeast MEP2 promoter activation.** MEP2 promoter-dependent lacZ expression was determined using a beta-galactosidase assay. At the indicated time points, 1.5 $OD_{600}$ units were harvested, washed once with water and lysed for 30 s in 380 µl Z-buffer (100 mM sodium phosphate buffer pH 7.0, 10 mM KCl, 1 mM $MgSO_4$, 40 mM 2-mecaptoethanol) with 50 µl 0.1% SDS and 50 µl $CHCl_3$ using a beadmill (Mini-Beadbeater). After addition of 50 µl 4 mg/ml o-nitrophenylgalactoside (Serva), suspensions were incubated at room temperature and stopped with 125 µl 1 M $Na_2CO_3$ when the suspensions turned yellow. Samples were centrifuged for 5 min at $2700 \times g$ and absorption of the supernatant was measured at 450 nm. To

assess the fraction of live cells, aliquots of the cultures were stained with PI and cell death measured by flow cytometry. Miller units/fraction alive were calculated using the formula $[1000x\ A_{450}]/[\text{volume in ml}*OD_{600}*\text{incubation time in min*fraction PI-negative cells}]$.

**Determination of yeast metabolomic changes.** Yeast cells cultured in deep well plates were treated with 100 µM DMC (or 0.2% DMSO as control) for 72 h. For each replicate, 5 ml culture were rapidly harvested by microfiltration using 0.45 µm PVDF-filters, immediately washed with 10 ml ultrapure water, snap-frozen in liquid nitrogen, and kept at −80 °C prior to extraction. Boiling ethanol (BE) extraction was performed with 2.5 ml pre-heated BE buffer (75% v/v ethanol, 15 mM ammonium acetate pH 7.5) for 2 min at 96 °C in a water bath with short vortexing every 30 s. After extraction, the cells' remnants were pelleted by centrifugation for 3 min at $2500 \times g$ at −20 °C, the supernatant was concentrated by nitrogen evaporation and dry-frozen. Samples were dissolved in 100 µl LCMS-$H_2O$, centrifuged at $17,000 \times g$ for 5 min and the supernatant used for LC/MS.

All samples were measured with a LC/MS system from Thermo Fisher Scientific™. A Dionex Ultimate 3000 HPLC setup equipped with an Atlantis T3 C18 analytical column (Waters, USA) was used for compound separation prior to mass spectrometric detection with an Exactive™ Orbitrap system. A reversed-phase ion-pairing HPLC method was used for metabolite separation (adapted from ref. [61]). Tributylamine was used as ion-pairing agent. A 40 min gradient was applied and 2-propanol and an aqueous phase (5% methanol (v/v), 10 mM tributylamine, 15 mM acetic acid, pH 4.95) were used as eluent A and B, respectively (Supplementary Table 6). The injection volume was 10 µl per sample and an injection loop of 20 µl was used.

Negative ionisation of metabolites was carried out via heated electrospray ionization (HESI) prior mass spectrometric analysis. For the online detection of the analytes a full scan of all masses between 70 and 1100 $m/z$ with a resolution (R) of 50,000 (at $m/z$ 200) was used.

LC/MS-data acquisition was conducted with Xcalibur software (version 2.2 SP1, Thermo Fisher Scientific (Waltham, USA)), Raw data were converted into mzXML by msConvert (ProteoWizard Toolkit v3.0.5), and metabolites were targeted-searched using the in-house developed tool PeakScout[62]. Pure analytes were run on the same system to obtain exact reference retention times and fragmentation spectra. Raw data was further assessed with Microsoft Excel 2010. For metabolite clustering, metabolite areas were normalised to the cumulative signal of all metabolite areas for each day (the mean signal of each metabolite across all samples was set to 1) and log2 transformed. PCA analyses were performed using the tool Genesis 1.7.7 (Bioinformatics, Technical University of Graz). The complete data set is available in Supplementary Data File 1.

**Determination of yeast proteome using SILAC.** SILAC (stable isotope labeling by/with amino acids in cell culture) experiments were performed following previously published protocols[17]. Briefly, proteins from labeled yeast cells (Lys0 + Arg0 or Lys4 + Arg10) treated with 100 µM DMC for 24 h or 72 h were extracted by glass bead disruption in buffer P (50 mM Tris/HCl pH 7.4, 1% Triton X-100, 150 mM NaCl, 1 mM EDTA) containing complete® protease inhibitor cocktail (Roche), 1 mM PMSF (Sigma) and HDAC inhibitors trichostatin A (2 µM, Sigma) and nicotinamide (30 mM, Sigma). Protein concentration was determined using Bio-Rad protein assay (Bio-Rad) and 500 µg from each heavy and light extracts were mixed and stored at −80 °C prior to MS measurement. For MS sample preparation, probes were reduced with 1 mM DTT (Sigma-Aldrich) and alkylated using 5.5 mM iodoacetamide (Sigma-Aldrich). Proteins were separated by SDS-PAGE and digested in gel using trypsin (Promega) at 37 °C over night and the resulting peptide mixtures were processed on STAGE tips[63].

Mass spectrometric measurements were performed on a LTQ Orbitrap XL mass spectrometer (Thermo Fisher Scientific, Bremen, Germany) coupled to an Agilent 1200 nanoflow-HPLC (Agilent Technologies GmbH, Waldbronn, Germany). HPLC-column tips (fused silica) with 75 µm inner diameter (New Objective, Woburn, MA, USA) were self-packed with Reprosil-Pur 120 ODS-3 (Dr. Maisch, Ammerbuch, Germany) to a length of 20 cm. Samples were applied directly onto the column without pre-column. A gradient of A [0.5% acetic acid (high purity, LGC Promochem, Wesel, Germany) in water (HPLC gradient grade, Mallinckrodt Baker B.V., Deventer, Netherlands)] and B [0.5% acetic acid in 80% ACN (LC-MS grade, Wako, Germany) in water] with increasing organic proportion was used for peptide separation (loading of sample with 2% B; separation ramp: from 10% to 30% B within 80 min). The flow rate was 250 nl/min and for sample application 500 nl/min. The mass spectrometer was operated in the data-dependent mode and switched automatically between MS (max. of $1 \times 10^6$ ions) and MS/MS. Each MS scan was followed by a maximum of five MS/MS scans in the linear ion trap using normalised collision energy of 35% and a target value of 5000. Parent ions with a charge state from $z = 1$ and unassigned charge states were excluded for fragmentation. The mass range for MS was $m/z = 370$ to 2000. The resolution was set to 60,000. Mass-spectrometric parameters were as follows: spray voltage 2.3 kV; no sheath and auxiliary gas flow; ion-transfer tube temperature 125 °C.

The MS raw data files were uploaded into the MaxQuant software version 1.4.1.2[64], which performs peak detection, label-free quantification, and generates peak lists of mass error corrected peptides using the following parameters: carbamidomethylcysteine was set as fixed modification, methionine oxidation and

protein amino-terminal acetylation were set as variable modifications. Three miss cleavages were allowed, enzyme specificity was trypsin/P, and the MS/MS tolerance was set to 0.5 Da. Peak lists were searched by Andromeda for peptide identification using a Uniprot *Saccharomyces cerevisiae* database containing common contaminants such as keratins and enzymes used for in-gel digestion. Peptide lists were further used by MaxQuant to identify and relatively quantify proteins using the following parameters: peptide, and protein false discovery rates were set to 0.01, maximum peptide posterior error probability (PEP) was set to 1, minimum peptide length was set to 7, the PEP was based on Andromeda score, minimum number peptides for identification and quantitation of proteins was set to one and must be unique, and identified proteins were re-quantified.

For enrichment analysis, the DAVID Bioinformatics Resources 6.7 software, NIAID/NIH [https://david-d.ncifcrf.gov/], was used employing the following settings: functional annotation clustering (GOTERM_BP_FAT), medium stringency, Kappa similarity (term overlap: 3, similarity threshold: 0.5), classification (initial group membership: 3, final group membership: 3, multiple linkage threshold: 0.5), enrichment Threshold (EASE: 1.0) The reference proteome was set to all measured proteins at a given timepoint. For *P*-value calculation, Benjamini correction was used. The complete data set is available in Supplementary Data File 2.

The mass spectrometry proteomics data have been deposited to the ProteomeXchange Consortium via the PRIDE[65] partner repository with the dataset identifier PXD012108 and the project name: "The flavonoid 4,4′-dimethoxychalcone promotes autophagy-dependent longevity across species".

**_C. elegans_ strains and maintenance.** We followed standard nematode culture conditions[66]. Nematode rearing temperature was kept at 20 °C, and animals were maintained at 20 °C on Nematode Growth Media (NGM) agar supplemented with *Escherichia coli* (OP50 or transformed HT115). We employed the N2 (wild type) and MH4799:elt-1(ku491) IV strains for lifespan assays and the VK1093:vkEx1093 [P$_{nhx-2}$::mCherry::lgg-1] strain for autophagy measurements.

**_C. elegans_ lifespan assays.** DMC-treatment of *C. elegans* was carried out on NGM agar plates continuously during the entire lifetime starting as L1 larvae, unless otherwise specified. All agar plates were prepared from the same batch of NGM agar and seeded with dried bacteria (OP50), which were previously UV-killed to avoid interference with the xenobiotic-metabolising activity of *Escherichia coli*. DMC (41.6 μM) or an equivalent volume of the solvent DMSO was spotted on dried bacteria in treatment and control plates, respectively. Spotted plates were dried overnight at room temperature prior to use. The procedure was repeated each time worms were transferred to fresh plates. For RNAi lifespan experiments, RNAi feeding started at L4 larval stage. Worms were placed on NGM plates containing 0.5–1 mM IPTG and seeded with HT115 bacteria transformed with either the pL4440 vector or the test RNAi plasmid. The dsRNA-transformed bacteria atg-5 (Y71G12B.12), bec-1 (T19E7.3) and elt-1 (W09C2.1) were derived from the Ahringer *C. elegans* RNAi library[67]. All dsRNA bacterial clones were used at a concentration of 0.9 OD diluted with empty vector-harbouring bacteria, according to previous studies[68].

Survival analysis started from hatching, unless otherwise specified, and was carried out at 20 °C using synchronous populations of 60/80 animals per condition. Animals were scored as dead or alive and transferred every day on fresh plates during the fertile period, and then every other day or every 3 days until death. Worms were considered dead when they stopped pharyngeal pumping and responding to touch. Worms that died because of internal bagging, desiccation due to crawling on the edge of the plates, or gonad extrusion were scored as censored. These animals were included in lifespan analyses up to the point of censorship. Note that due to the rather high biological variance associated with ageing experiments, replicates of survival assays are shown as separate experiments; for details, see Supplementary Table 2.

**_C. elegans_ autophagy measurements.** To quantify autophagy induction, the VK1093:vkEx1093 [P$_{nhx-2}$::mCherry::lgg-1] strain was used. Worms were treated with 41.6 μM DMC from the L1 stage for 48 h. For silencing experiments, worms were transferred as L4 on the respective plates (control and corresponding RNAi ± DMC). After 24 h of silencing the (adult) worms were acquired using a fluorescence microscope.

Using ImageJ, [http://imagej.nih.gov/ij/], the pictured worms were manually selected and the number of foci counted, using the automated function in ImageJ "Analyze Particles" on the adjusted images. To avoid artifacts, all the pictures were acquired and adjusted using the same settings.

**_D. melanogaster_ strains and media.** We used *Drosophila melanogaster* w[1118] or Atg7[−/−] flies. The latter were generated by crossing Atg7[d77/+] virgins with Atg7[d14/+] male flies[69], both isogenised to w[1118]. Fly experiments were performed using two different media: (i) CSY, containing 1% agar (BD) 1% Bacto™ yeast extract (BD), 5% sucrose (Roth), 0.8% cornmeal, 0.03% (v/v) propionic acid and 0.3% (v/v) 4-hydroxybenzoate solution (Sigma-Aldrich, 100 g/l in 100% ethanol), or (ii) Bloomington cornmeal-molasses medium, containing 0.8% agar, 0.75% dry baker's yeast, 0.83% soy meal, 8.5% sugar beet syrup, 6.7% corn meal, 0.0525% (v/

v) propionic acid and 0.42% 4-hydroxybenzoate solution (310 g/l in 100% ethanol). For both media, propionic acid and 4-hydroxybenzoate were added when the food had cooled down to 60 °C. DMC was added at 40 °C at a final concentration of 200 μM and a DMSO concentration of 0.1%. Food was stored at 4 °C for no longer than 1 week.

**_D. melanogaster_ lifespan analyses.** Flies were reared in standard wide plastic vials with plugs and hatched for three days until they were collected, allowed to mate for 24 h and then sexed under CO$_2$ anaesthesia and sorted into portions of 20 flies with a maximal, total anaesthesia time of 6 min. For brain immunofluorescence, flies were anesthetised for no longer than 2 min. On CSY, flies were transferred to fresh food three times a week, on Bloomington cornmeal-molasses medium every other day. Accidentally escaped flies were censored. All ageing experiments were carried out in climate chambers at 25 °C and 70% humidity with a 12 h/12 h light/dark cycle. Note that due to the rather high biological variance associated with ageing experiments, replicates of survival assays are shown as separate experiments; for details, see Supplementary Table 2.

**Fly food consumption assay.** Feeding behaviour was determined by capillary feeding assays. Briefly, plastic fly vials were filled with 3 ml 1% agar and three small air holes were introduced into the vial wall with a hot syringe. Vials were sealed with Parafilm (Bemis NA) and rubber caps, containing two additional air holes for ventilation and two holes for 5 μl glass capillaries (Hirschmann). Four female flies per vial were accustomed to the feeding procedure for 2 days. Then, feeding of the test food (2.5% yeast extract, 2.5% sucrose and 200 μM DMC or respective amount of DMSO) was monitored over three consecutive 10–15 h intervals. Five independent vials per condition were analysed and feeding behaviour expressed as μl food*fly$^{-1}$*12 h$^{-1}$.

**Fly fecundity assay.** Flies (1–4 days old) were transferred to new vials and allowed to mate for 48 h before sorting them into portions of 10 female animals under CO$_2$ anaesthesia. Flies were aged on Bloomington food containing 200 μM DMC or control food. At the indicated time points, flies were transferred to fresh food for 16 h and subsequently first instar larvae and eggs were counted. In total, six independent vials per condition were analysed.

**Fly autophagy measurements.** w[1118] flies were reared on Bloomington cornmeal-molasses medium, containing 0.47% agar, 0.75% dry baker's yeast, 0.83% soy meal, 8.5% sugar beet syrup, 6.7% corn meal, 0.0525% (v/v) propionic acid and 0.42% 4-hydroxybenzoate solution (310 g/l in 100% ethanol) and allowed to hatch for 2 days. Then flies were transferred to new vials and allowed to mate for 24 h. Portions of 20 female flies were sorted under CO$_2$ anaesthesia with a maximal anaesthesia of 2 min and transferred to small vials containing control food (0.1% DMSO) or food supplemented with 200 μM DMC, both containing 1% agar. Flies were transferred to fresh food every other day and aged for 3 (young) or 30 days (old) at 25 °C and 70% humidity with a 12 h/12 h light/dark cycle.

Brains from 30-days old female flies were dissected and collected in ice cold haemolymph-like saline (HL-3) solution. Following fixation in phosphate buffered saline containing 0.7% Triton X-100 (PBS-T) and 4% paraformaldehyde for 30 min and several washing steps in PBS-T, unspecific antibody binding sites were blocked with 10% normal goat serum. Brains were incubated with a polyclonal anti-Ref(2)P (Rabbit, gift from Dr. Gabor Juhasz, 1:8000) antibody for 48 h, washed six times, and subsequently incubated with the corresponding Alexa 488-linked secondary antibody (Rabbit, Invitrogen, [#A11034], 1:500) for another 16 h. 11–12 brains of each condition were stored in VectaShield (Vector Laboratories LTD., Peterborough, United Kingdom) in glass-bottom petri dishes at 4 °C and scanned by confocal microscopy within 2–4 weeks.

Spatial image data was generated using a Leica SP5 confocal microscope with spectral detection (Leica Microsystems, Inc.) and a HCX IRAPO L *25x/0.95 NA* water immersion objective. Z-stacks were acquired using a resonant scanner (8000 Hz, 8× line averaging, 1024 × 1024 pixels). Alexa 488 fluorescence was excited at 488 nm and emission detected at 500–550 nm. Image noise was reduced by image deconvolution (Huygens Pro, SVI B.V.; optimised maximum-likelihood estimation approach; five iterations). Processed z-stacks were projected using the maximum-intensity projection method. Image analysis of projections was performed using ImageJ software, [http://rsbweb.nih.gov/ij/]. The central brain region of 11–12 brains per condition was selected manually with the free-hand tool, and the average grey value related to the selected area of each brain was measured.

**Cell culture conditions and cell lines.** Culture media and supplements for cell culture were purchased from Gibco-Invitrogen (Carlsbad, CA, USA) and plasticware from Corning (Corning, NY, USA). Human osteosarcoma U2OS cells (cell line was purchased from ATCC [HTB-96]) and their GFP-LC3-expressing derivatives were cultured in DMEM medium containing 10% foetal bovine serum, 100 mg/l sodium pyruvate, 10 mM HEPES buffer, 100 units/ml penicillin G sodium and 100 μg/ml streptomycin sulphate (37 °C, 5% CO$_2$). Human hepatocellular carcinoma HepG2 cells (cell line was purchased from ATCC [HB8065]) were cultured in EMEM medium supplemented with 10% fetal bovine serum, 100 mg/l sodium pyruvate, 10 mM HEPES buffer, 100 units/ml penicillin G sodium and 100

µg/ml streptomycin sulphate. Human colorectal cancer HCT116 cells (cell line was purchased from ATCC [CCL-247]) were cultured in McCoy's medium enriched with 10% fetal bovine serum, 100 mg/l sodium pyruvate, 10 mM HEPES buffer, 100 units/ml penicillin G sodium and 100 µg/ml streptomycin sulfate. For autophagy induction, cells were treated for 8 h with 50 µM DMC (Sigma Aldrich [S617237]) in presence or absence of 50 µM chloroquine (Sigma Aldrich [C6628]) to properly assess autophagic flux.

**Cell culture autophagy measurements by automated microscopy.** GFP-LC3 U2OS cells were generated by transfection of U2OS cells with pEGFP-LC3 plasmid and were maintained by selection with Neomycin. GFP-LC3 HCT116 cells were generated by transfection with lentiviral GFP-LC3 construct (Millipore [17-10193]). U2OS or HCT116 cells stably expressing GFP-LC3 were seeded in 96-well or 384-well imageing plates (Greiner) 24 h before stimulation. Cells were treated with DMC for 8 h, in presence or absence of Chloroquine (for 2 h before fixation). Subsequently, cells were fixed with 4% PFA and counterstained with 10 µM Hoechst 33342. Images were acquired using a BD pathway 855 automated microscope (BD Imageing Systems, San José,USA) equipped with a 40X objective (Olympus, Center Valley, USA) coupled to a robotised Twister II plate handler (Caliper Life Sciences, Hopkinton, USA). Images were analysed for the presence of GFP-LC3 dots in the cytoplasm by means of the BD Attovision software (BD Imageing Systems). Cell surfaces were segmented and divided into cytoplasmic and nuclear regions according to manufacturer standard proceedings. RB 2x2 and Marr–Hildreth algorithms were used to recognise cytoplasmic GFP-LC3 positive dots. For siRNA analyses, U2OS cells stably expressing GFP-LC3 were transfected with scramble siRNA (siCtr) or siRNAs targeting GATA1 (siGATA1), GATA2 (siGATA2), GATA3 (siGATA3), GATA4 (siGATA4), GATA5 (siGATA5), GATA6 (siGATA6), TRPS1 (siTRPS1) and Atg5 (siAtg5) (3 individual siRNA sequences per each gene, see Table 5) for 48 h, followed by 50 µM DMC (or 10 µM rapamycin) or being kept in control condition for additional 6 h. Thereafter, cells were fixed and autophagy measured as described above. For quantification, the mean value and SEM from the data of all three individual siRNA sequences were calculated. Knockdown efficiency was verified by western blot analysis (Supplementary Figure 13).

**Yeast-like chronological senescence assay.** The assay was performed as described in Leontieva et al[70]. In brief, 80,000 cells were seeded into 96-well plates and left untreated or treated with 50 µM DMC. After 5 days, dead cells and conditioning media were removed, cells were trypsinised and a 10% aliquot was plated in fresh medium-filled six-well plates. After 1 week, clones were marked trough crystal violet staining and counted.

**Determination of S6K1 phosphorylation.** U20S cells were treated for 6 h with two different doses of DMC or rapamycin (1 µM). mTORC1 activity was assessed by immunoblotting and evaluated as levels of phosphorylation of its target ribosomal protein S6 kinase beta-1 (S6K1/P70S6K1) on threonine 389 using a phospho-specific antibody (Cell Signalling Technology, [#9205]). Rapamycin was used as a positive control of mTORC1 inhibition.

**Plant verification and extraction.** *Angelica keiskei koidzumi* plants were purchased from a local retailer. The plants displayed the morphological characteristics for the species, including pinnate leaves, serrate, 3-parted leaflets, long petioles with sheathing and the characteristic colour of its sap. In addition, plant DNA extracted from snap-frozen leaves was genotypically verified. Therefore, nuclear rDNA (internal transcribed spacer 1, 5.8S ribosomal RNA gene and internal transcribed spacer 2) was amplified with primers ITS4 and ITS5 (Supplementary Table 5) and the product sequenced using the same primers (MWG Biotech, Ebersberg, Germany). Comparison to GenBank accession no. GU395158.1 confirmed the genotype for *Angelica keiskei koidzumi*.

For detection of DMC in *Angelica keiskei koidzumi*, fresh plant material (~2 g of roots and stipes/leaves, respectively) was harvested, cut in 5 × 5 mm pieces, snap-frozen and pulverised with pre-cooled steel beads in a bead mill (mini-beadbeater, Biospec Products). Plant powder was dry-frozen and kept at −80 °C for at least 24 h prior to extraction (for extraction procedure, refer to "Extractions and measurements of DMC").

**Mouse husbandry and serum preparation.** Male C57BL/6 mice were purchased from Janvier Labs, France. Treatment of animals started at an age of 12 months. All mice were kept and treated in accordance with national and European ethical regulations (Directive 2010/63/EU) and the experiments approved by the responsible government agency (Bundesministerium für Wissenschaft, Forschung und Wirtschaft, BMWFW, Austria). Mice were fed ad libitum with regular food (pellets, Ssniff, Germany) supplemented with 0.25% DMC for 7 days. Animal physical activity, exploration, defecation, food and water intake as well as body weight was carefully monitored to ensure normal behaviour. At the end of the experiment, the animals were anaesthetised by isoflurane inhalation and sacrificed. Plasma was obtained by centrifugation at 200 × g for 10 min at 4 °C and kept at −80 °C prior to extraction.

For mouse autophagy measurements, 6-week-old male wild type C57BL/6 or Atg4b−/− mice (Harlan, France) were used. Animal experiments were in compliance with the EU Directive 63/2010 and protocol APAFIS #10511-2017070511526660 v2 was approved by the Ethical Committee of the CRC (CEEA no. 5, registered at the French Ministry of Research). Atg4b−/− mice were kindly provided by Dr. Carlos Lopez-Otin (Oviedo University, Spain).

Mice were housed in a temperature-controlled environment with 12 h light/dark cycles and received food and water ad libitum. Mice were injected intraperitoneally with a single 100 mg/kg dose of DMC, dissolved in 50 µl DMSO, and 30 mg/kg of Leupeptin (2 h prior to sacrifice) and 6 h later were sacrificed. Tissues were snap-frozen in liquid nitrogen and homogenised (two cycles of 20 s at 5500 rpm [motor speed]) using Precellys 24 tissue homogenator (Bertin Technologies, Montigny-le-Bretonneux, France) in a 20 mM Tris buffer (pH 7.4) containing 150 mM NaCl, 1% Triton X-100, 10 mM EDTA and Complete® protease inhibitor cocktail (Roche Applied Science, Penzberg, Germany). Tissue extracts were then centrifuged at 12,000 × g at 4 °C and supernatants were collected. Protein concentration in the supernatants was evaluated by the bicinchoninic acid technique (BCA protein assay kit, Pierce Biotechnology, Rockford, USA).

For prolonged ischemia in vivo, 3-month-old male C57BL/6J wild type or Atg7 cardiac-specific knockout (Atg7cKO) mice were obtained from the Jackson Laboratory. All mice were kept and treated in accordance with national ethical regulations (IACUC protocol #17022) and the experiments approved by the responsible institutional agency (Rutgers-New Jersey Medical School's Institutional Animal Care and Use Committee). Mice were housed in a temperature-controlled environment with 12-h light/dark cycles where they received food and water ad libitum. Three-month-old C57BL/6 J mice were anesthetised by intraperitoneal injection of pentobarbital sodium (60 mg/kg). A rodent ventilator (Minivent; Harvard Apparatus Inc.) was used with 65% oxygen. The animals were kept warm using heat lamps. Rectal temperature was monitored and maintained between 36.8 and 37.2 °C. The chest was opened by a horizontal incision at the fourth intercostal space. Ischemia was achieved by ligating the anterior descending branch of the left coronary artery (LAD) using an 8-0 prolene suture 2 mm below the border between the left atrium and LV. Ischemia was confirmed by ECG change (ST elevation) and colour change of myocardium. When recovered from anaesthesia, the mice were returned to their cages. They were housed in a climate-controlled environment.

**Determination of metabolic changes in mice heart and liver.** Animals were treated as described for mouse autophagy measurements (see "Mouse husbandry and serum preparation" section). All standard and reactives used were from Sigma-Aldrich except ammonium carbonate (VWR). For sample preparation of liver and heart tissue, about 30 mg of tissues for each condition were first weighted and solubilised into 1.5 ml polypropylene microcentrifuge tubes with ceramic beads with 1 ml of cold lysate buffer (MeOH/Water/Chloroform, 9/1/1, −20 °C). They were then homogenised three times for 20 s at 5,500 rpm (motor speed) using Precellys 24 tissue homogenator (Bertin Technologies, Montigny-le-Bretonneux, France), followed by a centrifugation (10 min at 15000 × g, 4 °C). Then upper phase of the supernatant was split in two parts: the first 270 µl used for the Gas Chromatography coupled by Mass Spectrometry (GC/MS) experiment in microtubes centrifugation, the others 250 µl used for the Ultra High Pressure Liquid Chromatography coupled by Mass Spectrometry (UHPLC/MS) experimentations.

Concerning the GC-MS aliquots, 150 µl were transferred from the microtube centrifugation to a glass tube and evaporated. 50 µl of methoxyamine (20 mg/ml in pyridine) was added on dried extracts, then stored at room temperature in dark, for 16 h. The day after, 80 µl of MSTFA was added and final derivatization occurred at 40 °C for 30 min. Samples were then transferred in vials and directly injected into GC-MS.

Concerning the LC-MS aliquots, the collected supernatant was evaporated in microcentrifuge tubes at 40 °C in a pneumatically-assisted concentrator (Techne DB3, *Staffordshire, UK*). The LC-MS dried extracts were solubilised with 450 µl of MilliQ water and aliquoted in three microcentrifuge tubes (100 µl) for each LC method and one microcentrifuge tube for safety.

Aliquots for analysis were transferred in LC vials and injected into LC/MS or kept at −80 °C until injection.

For sample preparation of polyamines (liver, heart), about 30 mg of tissues for each condition were first weighted and solubilised into 1.5 ml polypropylene microcentrifuge tubes, with 1 ml of cold lysate buffer with 1% sulfosalicylic acid (MeOH /water 1% SSA, 9/1, −20 °C). They were then homogenised three times for 20 s at 5500 rpm (motor speed) using Precellys 24 tissue homogenator (Bertin Technologies, Montigny-le-Bretonneux, France), followed by a centrifugation (10 min at 15,000 g, 4 °C). Six hundred microlitresof the upper phase of the supernatant was collected and evaporated in microcentrifuge tubes at 40 °C in a pneumatically-assisted concentrator (Techne DB3, Staffordshire, UK). The LC-MS dried extracts were solubilised with 300 µl of MilliQ water, centrifuged (10 min at 15,000 × g, 4 °C), and 50 µl were transferred in polypropylene vial injection for LC method and the rest was transferred in microcentrifuge tube for safety. Aliquots transferred in polypropylene vials were injected into LC/MS or kept at −80 °C until injection.

A number of targeted and untargeted analyses were performed to obtain a comprehensive picture of DMC's metabolomic impact. Note that all MRM transitions for the below methods are available upon request.

(1) Targeted analysis of CoAs and nucleoside phosphates by ion pairing ultra-high performance liquid chromatography (UHPLC) coupled to a Triple Quadrupole (QQQ) mass spectrometer was performed on a RRLC 1260 system (Agilent Technologies, Waldbronn, Germany) coupled to a Triple Quadrupole 6410 (Agilent Technologies) equipped with an electrospray source operating in positive mode. The gas temperature was set to 350 °C with a gas flow of 12 L/min. The capillary voltage was set to 3.5 kV. 10 µL of sample were injected on a Column XDB-C18 (100 mm × 2.1 mm particle size 1.8 µm) from Agilent technologies, protected by a guard column XDB-C18 (5 mm × 2.1 mm particle size 1.8 µm) and heated at 40 °C by a pelletier oven. Heat the column more than the room temperature allowed rigorous control of the column temperature. The gradient mobile phase consisted of water with 2 mM of DBAA (A) and acetonitrile (B). The flow rate was set to 0.2 ml/min, and gradient as follow: initial condition was 90% phase A and 10% phase B, maintained during 4 min. Molecules were then eluted using a gradient from 10% to 95% phase B over 3 min. The column was washed using 95% mobile phase B for 3 min and equilibrated using 10% mobile phase B for 3 min. The autosampler was kept at 4 °C. At the end of the batch of analysis, column was rinsed with 0.3 ml/min of MilliQ water (phase A) and acetonitrile (phase B) as follow: 10% phase B during 20 min, to 90% phase B in 20 min, and maintained during 20 min before shutdown. The collision gas was nitrogen. The scan mode used was the MRM for biological samples. Peak detection and integration of the 23 analytes were performed using the Agilent Mass Hunter quantitative software (B.07.01).

(2) Widely-targeted analysis of intracellular metabolites gas chromatography (GC) coupled to a triple quadrupole (QQQ) mass spectrometer was performed on a 7890 A gas chromatography (Agilent Technologies, Waldbronn, Germany) coupled to a triple quadrupole 7000 C (Agilent Technologies, Waldbronn, Germany) equipped with a High sensitivity electronic impact source (EI) operating in positive mode. The front inlet temperature was 250 °C, the injection was performed in splitless mode. The transfer line and the ion-source temperature were 250 °C and 230 °C, respectively. The septum purge flow was fixed at 3 mL/min, the purge flow to split vent operated at 80 mL/min during 1 min and gas saver mode was set to 15 ml/min after 5 min. The helium gas flowed through the column (J&WScientificHP-5MS, 30 m × 0.25 mm, i.d. 0.25 mm, d.f., Agilent Technologies Inc.) at 1 ml/min. Column temperature was held at 60 °C for 1 min, then raised to 210 °C (10 °C/min), followed by a step to 230 °C (5 °C/min) and reached 325 °C (15 °C/min), and be hold at this temperature for 5 min. The collision gas was nitrogen. The scan mode used was the MRM for biological samples. Peak detection and integration of the 133 analytes were performed using the Agilent Mass Hunter quantitative software (B.07.01).

(3) Targeted analysis of polyamines by ion pairing UHPLC coupled to a Triple Quadrupole (QQQ) mass spectrometer was performed on a RRLC 1260 system (Agilent Technologies, Waldbronn, Germany) coupled to a Triple Quadrupole 6410 (Agilent Technologies) equipped with an electrospray source operating in positive mode. The gas temperature was set to 350 °C with a gas flow of 12 L/min. The capillary voltage was set to 3.5 kV. 10 µL of sample were injected on a Column Kinetex C18 (150 mm × 2.1 mm particle size 2.6 µm) from Phenomenex, protected by a guard column C18 (5 mm × 2.1 mm) and heated at 40 °C by a pelletier oven. The gradient mobile phase consisted of water with 0, 1% of HFBA (A) and acetonitrile with 0,1% of HFBA (B) freshly made. The flow rate was set to 0.2 ml/min, and gradient as follow: initial condition was 95% phase A and 5% phase B. Molecules were then eluted using a gradient from 5% to 40% phase B over 10 min. The column was washed using 90% mobile phase B for 2.5 min and equilibrated using 5% mobile phase B for 4 min. The autosampler was kept at 4 °C. At the end of the batch of analysis, column was rinsed with 0.3 ml/min of MilliQ water (phase A) and acetonitrile (phase B) as follow: 10% phase B during 20 min, to 90% phase B in 20 min, and maintained during 20 min before shutdown. The collision gas was nitrogen. The scan mode used was the MRM for biological samples. Peak detection and integration of the 14 analytes were performed using the Agilent Mass Hunter quantitative software (B.07.01).

(4) Targeted analysis of bile acids by ion pairing UHPLC coupled to a Triple Quadrupole (QQQ) mass spectrometer was performed on a RRLC 1260 system (Agilent Technologies, Waldbronn, Germany) coupled to a Triple Quadrupole 6410 (Agilent Technologies) equipped with an electrospray source operating in positive mode. The gas temperature was set to 325 °C with a gas flow of 12 L/min. The capillary voltage was set to 4.5 kV. 10 µL of sample were injected on a Column Poroshell 120 EC-C8 (100 mm × 2.1 mm particle size 2.7 µm) from Agilent technologies, protected by a guard column XDB-C18 (5 mm × 2.1 mm particle size 1.8 µm) and heated at 40 °C by a pelletier oven. The gradient mobile phase consisted of water with 0.2% of formic acid (A) and acetonitrile/isopropanol (1/1; v/v) (B) freshly made. The flow rate was set to 0.3 ml/min, and gradient as follow: initial condition was 70% phase A and 30% phase B, maintained during 1.5 min. Molecules were then eluted using a gradient from 30% to 60% phase B over 9 min. The column was washed using 98% mobile phase B for 2 min and equilibrated using 30% mobile phase B for 2 min. After each injection, the needle was washed twice with isopropanol and thrice with water. The autosampler was kept at 4 °C. At the end of the batch of analysis, column was rinsed with 0.3 ml/min of MilliQ water (phase A) and acetonitrile (phase B) as follow: 10% phase B during 20 min, to 90% phase B in 20 min, and maintained during 20 min before shutdown. The collision gas was nitrogen. The scan mode used was the MRM for biological samples. Peak

detection and integration of the 28 analytes were performed using the Agilent Mass Hunter quantitative software (B.07.01).

(5) The untargeted analysis of intracellular metabolites by UHPLC coupled to a Q-Exactive mass spectrometer using the Reversed-phase acetonitrile method was performed on a Dionex Ultimate 3000 UHPLC system (Thermo Scientific) coupled to a Q-Exactive (Thermo Scientific) equipped with an electrospray source operating in both positive and negative mode and full scan mode from 100 to 1200 m/z. The Q-Exactive parameters were: sheath gas flow rate 50 au, auxiliary gas flow rate 10 au, spray voltage 4 kV, capillary temperature 300 °C, S-Lens RF level 55 V. 10 µL of sample were injected on a SB-Aq column (100 mm × 2.1 mm particle size 1.8 µm) from Agilent Technologies, protected by a guard column XDB-C18 (5 mm × 2.1 mm particle size 1.8 µm) and heated at 40 °C by a pelletier oven. The gradient mobile phase consisted of water with 0.2% of acetic acid (A) and acetonitrile (B). The flow rate was set to 0.5 ml/min. Initial condition was 98% phase A and 2% phase B. Molecules were then eluted using a gradient from 2% to 95% phase B in 8 min. The column was washed using 95% mobile phase B for 3 min and equilibrated using 2% mobile phase B for 3 min. The autosampler was kept at 4 °C. At the end of the batch of analysis, column was rinsed with 0.4 ml/min of MilliQ water (phase A) and acetonitrile (phase B) as follow: 10% phase B during 20 min, to 90% phase B in 20 min, and maintained during 20 min before shutdown. Peak detection and integration were performed using the Thermo Xcalibur quantitative software (3.1.).

(6) Untargeted analysis of intracellular metabolites by UHPLC coupled to a Q-Exactive mass spectrometer using the Reversed-phase methanol method was performed on a Dionex Ultimate 3000 UHPLC system (Thermo Scientific) coupled to a Q-Exactive (Thermo Scientific) equipped with an electrospray source operating in both positive and negative mode and full scan mode from 66 to 990 m/z. The Q-Exactive parameters were: sheath gas flow rate 50 au, auxiliary gas flow rate 10 au, spray voltage 4 kV, capillary temperature 300 °C, S-Lens RF level 55 V. Ten microlitres of sample were injected on an Eclipse Plus column (100 mm × 2.1 mm particle size 1.8 µm) from Agilent Technologies, protected by a guard column XDB-C18 (5 mm × 2.1 mm particle size 1.8 µm) and heated at 40 °C by a pelletier oven. The gradient mobile phase consisted of water with 0.05% of formic acid (A) and methanol with 0.05% of formic acid (B). The flow rate was set to 0.35 ml/min. Initial condition was 95% phase A and 5% phase B during 4 min. Molecules were then eluted using a gradient from 5% to 60% phase B in 13 min. The column was washed using 90% mobile phase B for 2.5 min and equilibrated using 5% mobile phase B for 7.5 min. The autosampler was kept at 4 °C. At the end of the batch of analysis, column was rinsed with 0.4 ml/min of MilliQ water (phase A) and acetonitrile (phase B) as follow: 10% phase B during 20 min, to 90% phase B in 20 min, and maintained during 20 min before shutdown. Peak detection and integration were performed using the Thermo Xcalibur quantitative software (3.1.).

(7) Untargeted analysis of intracellular metabolites by UHPLC coupled to a Q-Exactive mass spectrometer using the HILIC method was performed on a Dionex Ultimate 3000 UHPLC system (Thermo Scientific) coupled to a Q-Exactive (Thermo Scientific) equipped with an electrospray source operating in both positive and negative mode and full scan mode from 66 to 990 m/z. The Q-Exactive parameters were: sheath gas flow rate 50 au, auxiliary gas flow rate 10 au, spray voltage 4 kV, capillary temperature 300 °C, S-Lens RF level 55 V. The sample were diluted with a final volume of 50% acetonitrile, then 5 µl were injected on a SeQuant ZIC-pHILIC (150 mm × 2.1 mm particle size 5 µm) from Merck Millipore, protected by a guard column SeQuant ZIC-pHILIC (20 mm × 2.1 mm particle size 5µm) and heated at 40 °C by a pelletier oven. The gradient mobile phase consisted of water with 20 mM of ammonium carbonate (A) and acetonitrile (B). The flow rate was set to 0.3 ml/min. Initial condition was 5% phase A and 95% phase B during 1 min. Molecules were then eluted using a gradient as follow: 92% in 2 min, 86.5% in 0.5 min, 65% in 17.5 min of mobile phase B. The column was washed using 10% mobile phase B for 4.5 min and then equilibrated for the next analysis using 95% mobile phase B for 9 min. The autosampler was kept at 4 °C. At the end of the batch of analysis, column was rinsed with 0.3 ml/min of water 5 mM acetate ammonium (phase A) and acetonitrile (phase B) as follow: 50% phase B during 21 min, to 80% phase B in 21 min, and maintained during 18 min before shutdown. To remove residual salts from tubing, additional rinse of the autosampler (water/acetonitrile; 9/1; v/v) was added during 15 min, with the column off-line. Peak detection and integration were performed using the Thermo Xcalibur quantitative software (3.1.). The complete data set is available in Supplementary Data Files 3 and 4.

**Metabolite clustering**. Hierarchical clustering was performed using the MetaboAnalyst 3.0 suite[71] with the log2-normalised data as input using following parameters (unless stated otherwise): missing value estimation: skipped; data normalisation: none; data transformation: none; data scaling: none; distance measure: Pearson; clustering algorithm: Ward; default colour contrast; top 50 metabolites ranked by t-tests; data standardisation: autoscale features. Combined pathway topology and enrichment analysis was performed using the pathway analysis tool of MetaboAnalyst 3.0 with following parameters: missing value estimation: skipped; data normalization: none; data transformation: none; data scaling: none; pathway enrichment analysis: global test; pathway topology analysis: relative-betweenness centrality; reference: all compounds of the selected pathways. Pathways with impact "0" were omitted. As hexose-phosphate cannot be clearly

assigned to a specific metabolite, it was excluded for yeast pathway topology analysis. The complete pathway topology analyses are available in Supplementary Data File 1 (yeast), Supplementary Data File 3 (mouse heart) and Supplementary Data File 4 (mouse liver).

**Measurement of infarct size**. Three hours after LAD ligation, the animals were anesthetised and intubated, and the chest was opened. To demarcate the ischaemic area-at-risk (AAR), Alcian blue dye (1%) was perfused into the aorta and coronary arteries. Hearts were excised, and LVs were sliced into 1-mm-thick cross sections. The heart sections were then incubated with a 1% triphenyltetrazolium chloride solution at 37 °C for 15 min. The infarct area (white), the AAR (not blue), and the total LV area from both sides of each section were measured using ImageJ software and the values obtained were averaged. The percentage of area of infarction and AAR of each section were multiplied by the weight of the section and then totaled from all sections. AAR/LV and infarct area/AAR were expressed as percentages.

**Ethanol intoxication and determination of ALT activity**. Mice were fasted for 6 h and then they were administered a 33% (vol/vol) ethanol solution at a total cumulative dosage of 4.5-g/kg by four equally divided gavages in 20-minute intervals, as described in Ding et al.[72] Control mice received the same volume of water. Vehicle (DMSO) or DMC intraperitoneally injected to mice 30 min before ethanol administration. Sub-mandibular blood collection occurred 16 h after ethanol binge. Hepatic damage was quantified as serum ALT activity (Alanine Aminotransferase activity) by means of a specific kit (Alanine Transaminase Activity Assay Kit, Abcam [ab105134]). ALT activity was calculated as $[mU^*mg^{-1}]$ of pyruvate produced in the reaction.

**Extractions and measurements of DMC**. For DMC extraction of plant samples, 10 mg of plant powder were placed in 1 ml of ethylacetate and the samples extracted at room temperature for an hour in an ultrasonic bath followed by another hour on a shaker at 145 rpm (motor speed). Afterwards, the samples were centrifuged, the supernatant evaporated using nitrogen flow at room temperature and the resulting residue reconstituted in 100 µl 60% acetonitrile. For DMC extraction of mice plasma, 40 µl of plasma were precipitated with 150 µl of acetonitrile, vortexed, and centrifuged. The supernatant was evaporated at room temperature under nitrogen flow. The residue was dissolved in 50 µl of 60% acetonitrile.

All obtained samples were analysed using LC/MS. Experiments were carried out on an Ultimate 3000 System (Dionex, LCPackings, Vienna, Austria) coupled to a LTQ Orbitrap (ThermoFinnigan, Vienna, Austria), using an ESI ion source in positive mode. The system was controlled by Xcalibur Software 1.4. Chromatography was carried out on a Gemini C18, 20 x5 mm 3 µm (Phenomenex, Torrance, USA) column using 0.1% formic acid in water (A) and 0.1% formic acid in acetonitrile (B) as mobile phase. The separation was performed under gradient elution conditions: 60% B up to 90% B in 8.5 min, keeping 90% B constant for 3 min and re-equilibrating for 4 min. The flow rate was 200 µl/min at 30 °C, injection volume 20 µl. For the mass spectrometry measurements 2 scan events were monitored simultaneously: scan mode in the FTMS and production ion scan of 269 (CID 35) in the LTQ ion trap.

**Immunoblotting**. Preparation of yeast cell extracts for the EGFP-Atg8 autophagy assay and expression levels of HA-tagged Atg proteins were performed as described[17]. Therefore, three $OD_{600}$ units of cells (~$1 \times 10^8$ cells) were harvested by centrifugation at $10,000 \times g$ for 1 min, washed once with 500 µl $H_2O$ and resuspended in 300 µl lysis buffer (1.85 M NaOH, 7.5% 2-mercaptoethanol) and incubated for 10 min on ice. Then, 300 µl 55% trichloroacetic acid were added, samples were mixed briefly and incubated for another 10 min on ice. The precipitate was pelleted by centrifugation at $10,000 \times g$ at 4 °C for 10 min and the supernatant was removed (followed by a short spin to remove residual supernatant). The pellet was resuspended in 150 µl sample buffer (200 mM Tris/HCl pH 6.8, 2% SDS, 10% glycerol, 120 mM DTT, 0.004% bromophenol blue), placed in an ultrasound bath for 5 min and then boiled for 5 min at 95 °C. Before gel electrophoresis, samples were centrifuged at $10,000 \times g$ for 1 min.

Immunoblotting was performed using standard protocols. Blots were probed with murine monoclonal antibodies against GFP (Roche, [#1814460], 1:5,000 in TBS + 1% dry milk) or HA (Sigma-Aldrich, [H-9658], 1:10,000 in TBS + 1% dry milk) and the respective peroxidase-conjugated affinity-purified secondary antibody (Sigma, [#F-9137], 1:10,000 in TBS + 1% dry milk).

For human cell culture experiments, proteins were extracted from cells by means of Radio Immunoprecipitation Assay (RIPA) buffer. Twenty-five micrograms of proteins were separated on 4–12% bis-tris acrylamide (Thermo Fisher Scientific Inc.) and electrotransferred to Immun-Blot®PVDF Membrane (Biorad) Membranes were then sliced horizontally in different parts according to the molecular weight of the protein of interest to allow simultaneous detection of different antigens within the same experiment. Unspecific binding sites were saturated by incubating membranes for 1 h in 0.05% Tween 20 (v:v in TBS) supplemented with 5% non-fat powdered milk (w:v in TBS), followed by an overnight incubation with primary antibodies specific for LC3B (Cell Signalling Technology, [#2775], rabbit, 1:1,000) and SQSTM1/p62 (Abnova, clone 2C11,

[#H00008878-M01], mouse, 1:10,000). Development was performed with appropriate horseradish peroxidase (HRP)-labelled secondary antibodies (Rabbit-HRP, Thermo Scientific, [#31460], 1:5,000; Mouse-HRP, Thermo Scientific, [#31430], 1:5,000) plus the Super Signal West Pico chemiluminescent substrate (Thermo Fisher Scientific Inc.). An anti-glyceraldehyde-3-phosphate dehydrogenase antibody was used to control equal loading of lanes (for yeast experiments: the antibody was a gift from Dr. Guenther Daum, rabbit, 1:40,000; for human cell and mouse experiments: Cell Signalling Technology, clone D16H11, [#5174], rabbit, 1:10,000). For knockdown efficiency tests of GATA siRNA and controls (Supplementary Figure 13), the following antibodies were used: Thermo Fisher Scientific #PA1099X, rabbit, 1:500 (GATA1), Thermo Fisher Scientific [#710242], rabbit, 1:100 (GATA2), Cell Signalling Technology [#5852], rabbit, 1:1000 (GATA3), Cell Signalling Technology [#36966], rabbit, 1:1000 (GATA4), Thermo Fisher Scientific [#PA547262], goat, 1:200 (GATA5), Cell Signalling Technology [#5851], rabbit, 1:1000 (GATA6), Abcam [#ab48820], rabbit, 1:2000 (TRPS1), Cell Signalling Technology [#12994], rabbit, 1:1000 (Atg5). Full scans of all immunoblots are depicted in Supplementary Fig. 17.

**Statistical analyses**. The number of independent biological replicates (n) is indicated in the figure legends of the corresponding graphs. If not stated otherwise, independence is defined for yeast experiments as different clones or transformants, aged separately; for cell culture as separately treated cultures; for C. elegans, Drosophila and mouse experiments as different animals. Sample size was pre-estimated based on the effect size observed in initial yeast experiments. There was no blinding in the experiments. Where normality criteria were met (calculated by Kolmogorow-Smirnow and Bartlett's tests), P-values were calculated using Student's t-test (two-sided, unpaired) or an analysis of variance (ANOVA) approach (for more than two groups), followed by a Bonferroni post hoc test. Non-parametric data was analysed using a Kruskal–Wallis test with Dunn's multiple comparison correction. Where necessary, data was log-transformed to achieve normality and homogenous variances and significance assessed for the transformed data. For data with two variables (e.g. treatment and strain) a two-way ANOVA was performed, followed by testing simple main effects (Bonferroni-corrected multiple comparisons of different levels of each factor in case of significant main factor or interaction). Significance of yeast ageing survival data (factors treatment and time) was tested using repeated measures ANOVA (two-way mixed ANOVA) with time as the within-subjects factor. For survival analyses, log-rank (Mantel-Cox) tests were performed between DMC-treated and untreated conditions and the significance threshold was adapted according to the number of pairwise comparisons. If not stated otherwise, asterisks indicate significance: ***$P < 0.001$, **$P < 0.01$, *$P < 0.05$. Statistics were performed using Origin Pro 2008 (OriginLab).

**Reporting summary**. Further information on experimental design is available in the Nature Research Reporting Summary linked to this article.

## Data availability

The raw mass spectrometry proteomics data to the ProteomeXchange Consortium via the PRIDE partner repository; the project accession number is PXD012108. The metabolomics data are available at Mendeley Data, [https://doi.org/10.17632/83xw8jtgh2.1] X. In addition, all raw and processed LC–MS/MS metabolomics and proteomics data are available as Supplementary Data files 1-4. The source data underlying Fig. 1 (b-f, j), 2 (b, d-e, g, i), 3 (a, c), 4 (a), 5 (a-b, d-g), 6 (a-b, d) and 7 (c-d) as well as those underlying Supplementary Figures 1 (a-e), 2(a-d, e-f), 4 (a-b), 5 (d-e), 6 (b, d, f, h-k, m-o), 7 (a-b, h), 8 (a-e), 9 (a-e), 10 (a-g), 11 (a-e, g), 14 (a) and 16 (a) are provided as a Source Data file. All other data supporting the findings of this study are available from the corresponding authors on reasonable request. A reporting summary for this Article is available as a Supplementary Information file.

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

## Acknowledgements
F.M. and D.C-G. are grateful to the Austrian Science Fund FWF (SFB-LIPOTOX F3007&F3012, W1226, P29203, P29262), in particular for the project P27893 ("Pro-autophagic polyphenols and polyamines for longevity") and the Austrian Federal Ministry of Education, Science and Research and the University of Graz for grants "Unkonventionelle Forschung" and "flysleep" (BMWFW-80.109/0001-WF/V/3b/2015). J.T. is funded by the FWF (W 1226, DK Metabolic and Cardiovascular Disease) at the University of Graz. K.K. is a fellow of the Doctoral College "Metabolic and Cardiovascular Disease" (FWF W1226) and was funded by the University of Graz. We acknowledge support from NAWI Graz and the BioTechMed-Graz flagship project "EPIAge". G. K. is supported by the Ligue contre le Cancer (équipe labelisée); Agence National de la Recherche (ANR) – Projets blancs; ANR under the frame of E-Rare-2, the ERA-Net for Research on Rare Diseases; Association pour la recherche sur le cancer (ARC); Cancéropôle Ile-de-France; Institut National du Cancer (INCa); Inserm (HTE); Institut Universitaire de France; Fondation pour la Recherche Médicale (FRM); the European Commission (ArtForce); the European Research Council (ERC); Fondation Carrefour; the LeDucq Foundation; the LabEx Immuno-Oncology; the RHU Torino Lumière, the Seerave Foundation, the SIRIC Stratified Oncology Cell DNA Repair and Tumour Immune Elimination (SOCRATE); the SIRIC Cancer Research and Personalised Medicine (CARPEM); and the Paris Alliance of Cancer Research Institutes (PACRI). N.V. would like to thank the Deutsche Forschungsgemeinschaft (DFG) and the Bundesministerium für Bildung und Forschung (BMBF) for funding. We thank Dr. Stefan Benke and Lydia Opriessnig for technical assistance and Dr. Sabrina Büttner as well as Dr. Ulrich Stelzl and Katharina Radakovics for advice and discussions. We thank Dr. Tarek Moustafa for providing the rps6 mutant strain and Dr. Carlos Lopez-Otin (Oviedo University, Spain) for kindly providing Atg4b−/− mice. We further thank Dr. Bruno André (Université Libre de Bruxelles, Belgium) for providing the PMEP2-LacZ fusion plasmid. We thank Dr. Guenther Daum (Technical University of Graz, Austria) for providing the GAPDH antibody for yeast western blot analyses. We also thank Dr. Gabor Juhasz (Eötvös Loránd University, Hungary) for providing the anti-Ref(2)P antibody for corresponding measurements in fly brains.

## Author contributions
D.C.-G., A.Z., G.K. and F.M. designed the research; D.C.-G., A.Z., K.K., S.Sc., S.J.H., M.A. B., T.P., J.T., C.D. and C.R. performed the yeast experiments; F.P., G.C., F.C. and V.S. performed the mouse and cell culture experiments; S.Ma., A.S., and N.V. designed and performed the nematode experiments; A.Z., K.K., S.J.H., C.S., C.B.B., S.Me., H.W. and S.J.S. designed and performed the Drosophila experiments; G.T., R.R. and C.M. performed the metabolomics from yeast and measurements from Ashitaba extracts; Z.H. and J.D. performed the proteomics; J.N. and J.S. performed the ischemia experiments; F.P., S.D., N.B., F.A., O.K. and D.P.E. performed the mouse metabolomics. T.E., F.S., M.A., S.Se. and T.R.P. helped with experimental design; D.C-G. and A.Z. analysed most of the data and D.C.-G., A.Z., G.K. and F.M. wrote the manuscript.

## Additional information

**Competing interests:** D.C-G., G.K., O.K. and F.M. are the scientific co-founders of Samsara Therapeutics. The remaining authors declare no competing interests.

Didac Carmona-Gutierrez[1], Andreas Zimmermann[1,2], Katharina Kainz[1], Federico Pietrocola[3,4,5,6], Guo Chen[3,4,5,6], Silvia Maglioni[7], Alfonso Schiavi[7], Jihoon Nah[8], Sara Mertel[9], Christine B. Beuschel[9], Francesca Castoldi[3,4,5,6,10], Valentina Sica[3,4,5,6], Gert Trausinger[11], Reingard Raml[11], Cornelia Sommer[1], Sabrina Schroeder[1], Sebastian J. Hofer[1], Maria A. Bauer[1], Tobias Pendl[1], Jelena Tadic[1], Christopher Dammbrueck[1], Zehan Hu[12], Christoph Ruckenstuhl[1], Tobias Eisenberg[1], Sylvere Durand[3,4], Noélie Bossut[3,4], Fanny Aprahamian[3,4], Mahmoud Abdellatif[12], Simon Sedej[12,13], David P. Enot[3,4], Heimo Wolinski[1], Jörn Dengjel[14], Oliver Kepp[3,4,5,6], Christoph Magnes[11], Frank Sinner[2,11], Thomas R. Pieber[2,11], Junichi Sadoshima[8], Natascia Ventura[7,15], Stephan J. Sigrist[9,16], Guido Kroemer[3,4,5,6,17,18] & Frank Madeo[1,13]

[1]Institute of Molecular Biosciences, NAWI Graz, University of Graz, Graz 8010, Austria. [2]Division of Endocrinology and Diabetology, Department of Internal Medicine, Medical University of Graz, Graz 8036, Austria. [3]Equipe 11 labellisée Ligue contre le Cancer, Centre de Recherche des Cordeliers, INSERM U 1138, Paris, France. [4]Metabolomics and Cell Biology Platforms, Gustave Roussy Comprehensive Cancer Center, Villejuif, France. [5]Université Paris Descartes, Sorbonne Paris Cité, Paris, France. [6]Université Pierre et Marie Curie, Paris, France. [7]IUF - Leibniz Research Institute for

Environmental Medicine, Düsseldorf 40225, Germany. [8]Department of Cell Biology and Molecular Medicine, Rutgers New Jersey Medical School, Newark, NJ, USA. [9]Institute for Biology/Genetics, Freie Universität Berlin, Berlin 14195, Germany. [10]Sotio a.c, 17000 Prague, Czech Republic. [11]Joanneum Research Forschungsgesellschaft m.b.H., HEALTH, Institute for Biomedicine and Health Sciences, Graz 8010, Austria. [12]Department of Cardiology, Medical University of Graz, Graz 8036, Austria. [13]BioTechMed Graz, Graz 8010, Austria. [14]Department of Biology, Université de Fribourg, Chemin du Musée 10, 1700 Fribourg, Switzerland. [15]Institute for Clinical Chemistry and Laboratory Diagnostic, Medical Faculty of the Heinrich Heine University, Moorenstrasse 5, 40225 Düsseldorf, Germany. [16]NeuroCure, Charité, Berlin 10117, Germany. [17]Pôle de Biologie, Hôpital Européen Georges Pompidou, Paris, France. [18]Karolinska Institute, Department of Women's and Children's Health, Karolinska University Hospital, Stockholm, Sweden. These authors contributed equally: Didac Carmona-Gutierrez, Andreas Zimmermann.

