## [Peer Review File · Nature Communications]

Editorial Note: This manuscript has been previously reviewed at another journal that is not operating a transparent peer review scheme. This document only contains reviewer comments and rebuttal letters for versions considered at Nature Communications. Mentions of prior referee reports have been redacted.

REVIEWERS' COMMENTS:

Reviewer #1 (Remarks to the Author):

[Redacted] Despite some concerns over reconciling pieces of their data with the known literature (and at times with their own) I was generally positive about the initial submission. Given the extensive additional data, [Redacted] I now recommend this paper as acceptable for publication. The work is novel and timely, and in my opinion concerns of all three reviewers have been sufficiently addressed.

Reviewer #2 (Remarks to the Author):

The authors have done a very nice job of addressing my previous concerns. In my opinion, the revised manuscript is much clearer and thereby more impactful.

Reviewer #3 (Remarks to the Author):

Although Flavanoids are known to have beneficial effects, DMC was reported as superior in reducing chronological age-related cell death when compared with resveratrol, rapamycin and other flavonoids. The authors have performed additional analyses and extended their mechanistic understanding of the role of autophagy in the DMC mechanism of action. The manuscript is substantially improved.

Reviewer #4 (Remarks to the Author):

The authors have responded in detail to my comments and have added substantial new data supporting their claims. I believe the manuscript can now be accepted for publication.

Point-by-point reply:

We thank the four reviewers for their time and thoughtful suggestions on our work throughout the peer-review process, which have clearly improved the manuscript. We are happy to read that our amendments incorporated in the revised version of our article are satisfactory, and that none of the reviewers has requested further modifications.

Reviewer #1 (Remarks to the Author):

[Redaction] Despite some concerns over reconciling pieces of their data with the known literature (and at times with their own) I was generally positive about the initial submission. Given the extensive additional data, [Redaction] I now recommend this paper as acceptable for publication. The work is novel and timely, and in my opinion concerns of all three reviewers have been sufficiently addressed.

Reviewer #2 (Remarks to the Author):

The authors have done a very nice job of addressing my previous concerns. In my opinion, the revised manuscript is much clearer and thereby more impactful.

Reviewer #3 (Remarks to the Author):

Although Flavanoids are known to have beneficial effects, DMC was reported as superior in reducing chronological age-related cell death when compared with resveratrol, rapamycin and other flavonoids. The authors have performed additional analyses and extended their mechanistic understanding of the role of autophagy in the DMC mechanism of action. The manuscript is substantially improved.

Reviewer #4 (Remarks to the Author):

The authors have responded in detail to my comments and have added substantial new data supporting their claims. I believe the manuscript can now be accepted for publication.